# The fossil insect assemblage associated with the Toarcian (Lower Jurassic) oceanic anoxic event from Alderton Hill, Gloucestershire, UK

**Emily J. Swaby**[1]*, **Angela L. Coe**[1], **Jörg Ansorge**[2], **Bryony A. Caswell**[3], **Scott A. L. Hayward**[4,5], **Luke Mander**[1], **Liadan G. Stevens**[6], **Aimee McArdle**[6]

1 School of Environment, Earth and Ecosystem Sciences, The Open University, Walton Hall, Milton Keynes, Buckinghamshire, United Kingdom, 2 Institut für Geologische Wissenschaften, Ernst-Moritz-Arndt-Universität Greifswald, Greifswald, Germany, 3 School of Environmental Science, Faculty of Science and Engineering, University of Hull, Hull, United Kingdom, 4 School of Biosciences, University of Birmingham, Birmingham, United Kingdom, 5 Birmingham Institute of Forest Research, University of Birmingham, Birmingham, United Kingdom, 6 Natural History Museum, London, United Kingdom

* emily.swaby@open.ac.uk

**Data Availability Statement:** All data files and photographs are available from the Dryas database at: https://doi.org/10.5061/dryad.wh70rxwv0.

## Abstract

Extreme global warming and environmental changes associated with the Toarcian (Lower Jurassic) Oceanic Anoxic Event (T-OAE, ~183 Mya) profoundly impacted marine organisms and terrestrial plants. Despite the exceptionally elevated abundances of fossil insects from strata of this age, only assemblages from Germany and Luxembourg have been studied in detail. Here, we focus on the insect assemblage found in strata recording the T-OAE at Alderton Hill, Gloucestershire, UK, where <15% of specimens have previously been described. We located all known fossil insects (n = 370) from Alderton Hill, and used these to create the first comprehensive taxonomic and taphonomic analysis of the entire assemblage. We show that a diverse palaeoentomofaunal assemblage is preserved, comprising 12 orders, 21 families, 23 genera and 21 species. Fossil disarticulation is consistent with insect decay studies. The number of orders is comparable with present-day assemblages from similar latitudes (30˚–40˚N), including the Azores, and suggests that the palaeoentomofauna reflects a life assemblage. At Alderton, Hemiptera, Coleoptera and Orthoptera are the commonest (56.1%) orders. The high abundance of Hemiptera (22.1%) and Orthoptera (13.4%) indicates well-vegetated islands, while floral changes related to the T-OAE may be responsible for hemipteran diversification. Predatory insects are relatively abundant (~10% of the total assemblage) and we hypothesise that the co-occurrence of fish and insects within the T-OAE represents a jubilee-like event. The marginally higher proportion of sclerotised taxa compared to present-day insect assemblages possibly indicates adaptation to environmental conditions or taphonomic bias. The coeval palaeoentomofauna from Strawberry Bank, Somerset is less diverse (9 orders, 12 families, 6 genera, 3 species) and is taphonomically biased. The Alderton Hill palaeoentomofauna is interpreted to be the best-preserved and most representative insect assemblage from Toarcian strata in the UK. This study provides an essential first step towards understanding the likely influence of the T-OAE on insects.

**Funding:** ES Grant number NE/S007350/1] Natural Environmental Research Council (NERC) through the Central England NERC Training Alliance (CENTA) Doctoral Training Partnership https://www.ukri.org/councils/nerc/ The sponsors did not play any role in the study design, data collection and analysis, decision to publish, or preparation of the manuscript.

**Competing interests:** The authors have declared that no competing interests exist.

## Introduction

Evidence for present-day anthropogenically-driven climate change is overwhelming [1], and contemporary global warming is significantly altering biodiversity and Earth's ecosystems at an unprecedented rate and scale [2]. Insects are the most affected groups of animals, with many insect taxa currently experiencing rapid declines [3]. Changes in climate greatly affect the physiology, diversity, reproduction and population size of insects [3–7]. However, despite the importance of insects within terrestrial ecosystems, research on how they are, and will, respond to a rapidly changing climate is still in its infancy [8, 9]. Accumulations of fossil insects provide opportunities to investigate the dynamics of insect diversity and ecology through time including how they have responded to previous environmental change. Here, we focus on a previously largely undescribed assemblage of fossil insects from the UK that is associated with the extreme climate change during the Toarcian Oceanic Anoxic Event (T-OAE), Early Jurassic (*c*. 183 Ma ago), to provide base-line information for further studies on the impact of this climatic event on insects.

It is now well established that the T-OAE was a sudden and severe global event of extreme environmental change, associated with widespread burial of marine carbon and a pronounced perturbation to the global carbon cycle [10–14]. Studies of records of the T-OAE have highlighted synchronous evidence for increased global palaeotemperatures [15–24], an enhanced hydrological cycle and global chemical weathering rates [25, 26], widespread marine anoxia [27] and increased atmospheric $CO_2$ levels [28].

The T-OAE had a profound influence on marine biota, including elevated mass extinction rates [29, 30], changes in biogeographical ranges of key taxa [31, 32], as well as changes in sea-floor community composition and the body size of marine organisms [32–37]. More recently it has been noted that on land, global warming, possible wildfires and acid rain associated with the T-OAE could have resulted in a decrease in land plant diversity [38], which then had significant effects on the rest of the trophic web including the extinction of diverse basal dinosaur families formerly referred to as 'prosauropods' [39]. Insects form a critical part of ecosystems, however their response to past periods of environmental change, including the T-OAE, has not previously been investigated in detail.

The insect material from the Toarcian (Lower Jurassic) strata of Western Europe reported to date is all preserved in marine deposits and is stratigraphically restricted to the lower Toarcian (*Cleviceras exaratum* ammonite Subzone, lower *Harpoceras falciferum* ammonite Zone), coinciding with the negative carbon isotope excursion that characterises the T-OAE [11]. This relationship, in addition to the lack of fossil insect accumulations in the strata above and below and their rarity in stratigraphical record in general, strongly suggests that their occurrence may be linked to palaeoenvironmental and palaeoecological conditions of the T-OAE. Abundant insect material associated with the T-OAE has been found in the UK (Alderton Hill, Gloucestershire [40], and Strawberry Bank, Somerset [41–43]), Northern Germany (e.g. Grimmen, Dobbertin and Braunschweig [44–47]), Southern Germany (e.g. Kerkhofen, Holzmaden and Mistelgau [48–50]) and Luxembourg (e.g. Bascharage and Sanem [51–57]). Smaller assemblages of Toarcian insect material have also been documented from Switzerland [58] and Belgium [59].

This study focuses on the early Toarcian succession of Alderton Hill, Gloucestershire, UK (SP006345) that yielded a rich and diverse insect fauna when it was exposed in the mid-19th century [60–63]. The succession is regarded as an important locality for Early Jurassic insect faunas [40] and has provided the type specimens of 30 Early Jurassic insect species. However, despite discovery more than 170 years ago, the specimens, now in museum collections, have only received limited scientific study, and a comprehensive synthesis and review of all insects

from the locality has yet to be published [40, 64]. Selected specimens from the Alderton Hill palaeoentomofauna were initially described during the late 19th century [65–67] and early 20th century [68–72]. In recent literature, only specimens from certain insect orders have been featured in publications, including: Odonata [73, 74], Blattodea [75], Diptera [76] and Necro-trichoptera [43, 77]. Most recently, Simms [64], based almost entirely on historical records [78, 79], provided a brief summary of the Alderton Hill succession, while Jarzembowski and Palmer [40] provided a list of the insect orders present based on the literature.

The only other known UK locality that has yielded a substantial amount of fossilized insects from the T-OAE is Strawberry Bank in Ilminster, Somerset (ST361148). This location has yielded approximately 800 individual insect fossils [41, 42, 63, 80], in addition to a diverse assemblage of marine nekton, including ichthyosaurs [81–83], crocodiles [80, 84], fishes [85], cephalopods [86], euarthropods [87], crustaceans [88], and molluscs [89]. The diverse fauna was exclusively collected by Charles Moore during the late 1840s [41, 63] from an exposure that is now inaccessible [82]. In 2019, a new temporary excavation of Strawberry Bank was undertaken by the Bath Royal Literary and Scientific Institution and the University of Bristol and research on this new exposure is on-going [89]. The original insect material from Straw-berry Bank collected by Moore was briefly mentioned by Brodie [63] but remained unde-scribed until Williams *et al.* [41] provided an overview.

Here, for the first time, we provide a detailed synthesis, description and analysis of the tax-onomy and taphonomy of all known fossil insect material from Alderton Hill, which we have located within UK museum and institution collections. We compare the Alderton Hill insect assemblage with the contemporaneous fossil insects from Strawberry Bank and present-day insect assemblages in order to investigate the taxonomic diversity, preservation and signifi-cance of the Toarcian Alderton Hill palaeoentomofauna. This study will provide an essential base-line for future studies assessing the impact of the T-OAE on insects and how global warming influences their critical ecological role within the biosphere.

## Location and geological setting

The Early Jurassic deposits that comprise Alderton Hill, Gloucestershire, UK were deposited within the Severn Basin, a north-south elongated basin extending approximately 50 km east-west and $\geq$ 90 km north-south [64], centred on what is now the town of Tewksbury. During the Early Toarcian, Gloucestershire was located at palaeolatitude of 35˚– 40˚N (Fig 1A), and the Severn Basin formed a small part of an extensive epicontinental shelf with several small islands (e.g. Lon-don-Brabant Massif, Cornubian Massif and the Welsh Massif; Fig 1B). The major landmasses of Laurentia and Eurasia were > 1000 km away (Fig 1B). Today, Alderton Hill is located approxi-mately 2 km southwest of the village of Dumbleton, Gloucestershire. Alderton Hill originally had three small quarries: Alderton Hill Quarry, Dumbleton Pit and Naunton Farm Quarry. However, the whole area is now extensively overgrown with almost no visible exposure. The sites of all three quarries are situated on an area of private land protected both as a geological Site of Special Scien-tific Interest (SSSI) and a Geological Conservation Review site (GCR) [90].

Stratigraphically, Alderton Hill comprises the Marlstone Rock Formation (Upper Pliensba-chian) overlain by the Dumbleton Member which forms the lowermost unit of the Whitby Mudstone Formation (Lower Toarcian) [64]. As the section has not been exposed for some time, we rely on the summary compiled by Simms [64] based on nineteenth century records (referenced therein) and Richardson [79]. Using this summary and our observations of the matrix around the fossil specimens, the Dumbleton Member is comprised of organic-rich, laminated blue-grey mudstones, with several bands of early diagenetic calcareous nodules, including the Fish Bed (Bed 10) that occurs *c.* 4 m above the base of the Dumbleton Member.

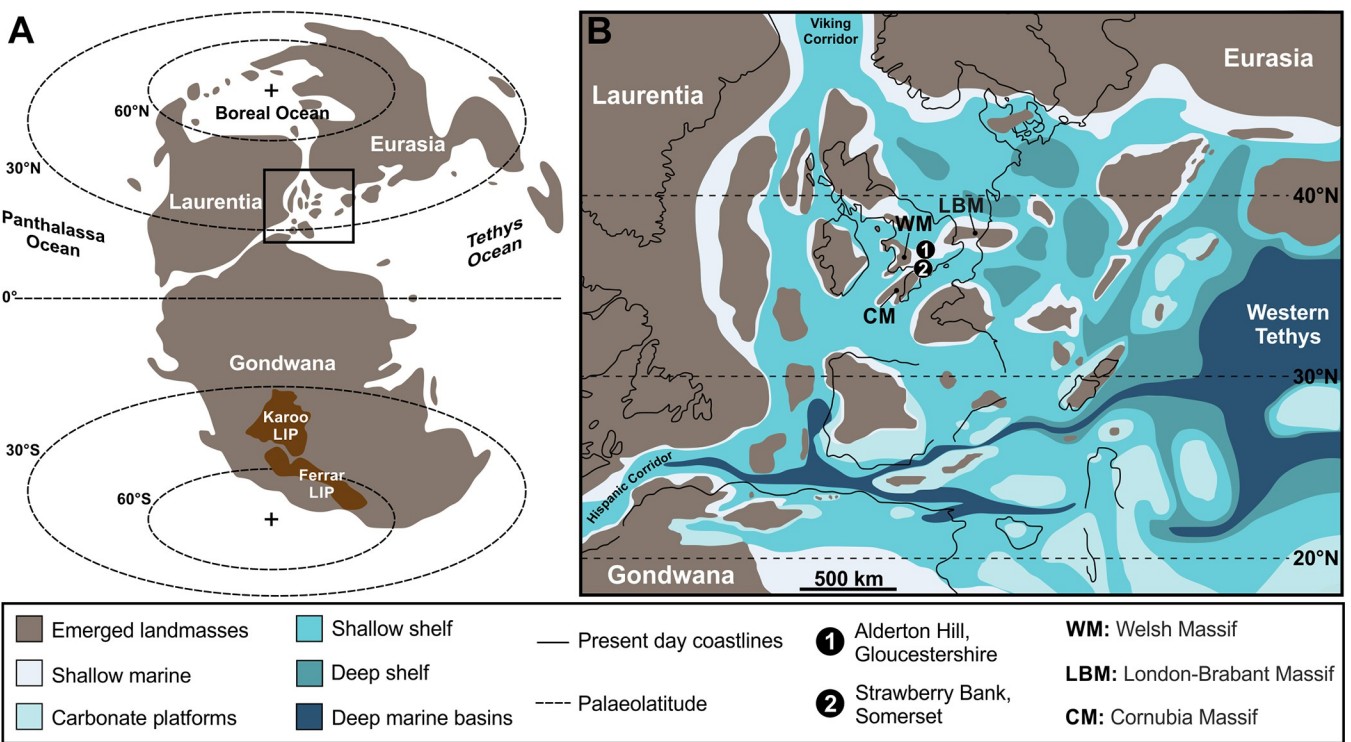

**Fig 1.** **(A)** Global palaeogeographical map of the Early Toarcian (modified from [22] under a CC BY license), showing the location of the Karoo and Ferrar large igneous provinces (LIPs) in South Gondwana [91]; the box shows B. **(B)** Palaeogeography of NW Tethys, modified from [92] under a CC BY license, showing the likely distribution of the Toarcian palaeo-landmasses superimposed on the present-day coastline, and the location of: (1) Alderton Hill, UK and (2) Strawberry Bank, UK.

The presence of the ammonites *Hildaites murleyi* (BGS GS.32040; Fig 2G), *H. forte* (BGS GS.32052; Fig 3 in plate 32 of [94]) and *Cleviceras elegans* within the Fish Bed indicate the horizon is situated within the *Cleviceras exaratum* Subzone, lower *Harpoceras falciferum* Zone [64, 93, 94]. This is stratigraphically comparable with the insect-bearing Saurian and Fish Zone of the Strawberry Bank Lagerstätte [41] (Fig 1B). In addition to the abundant insect material, the Fish Bed at Alderton Hill has yielded a high abundance of the small early teleost fish *Leptolepis coryphaenoides*, *L. normandica* and *L. concentricus* (Fig 2A–2C). During this study, a specimen assigned to *Pachycormus macropterus* (Fig 2D) was located in the Sedgwick Museum, Cambridge. There is also a marine invertebrate fauna dominated by nektonic and planktonic species, including teuthoids (Fig 2E, Fig 2I–2J) [95–97]; rare arthropods including the holotype of *Proeryon richardsoni* (Fig 2K) [98, 99]; molluscs, including the bivalve *Pseudomytiloides dubius* (Fig 2F) and an abundance of the small gastropod *Coelodiscus minutus* (Fig 2H) [40, 64]; together with rare reptile remains [81].

The other UK location that has yielded extensive insect fossils, Strawberry Bank, in the town Ilminster, Somerset, is located approximately 130 km southwest of Alderton Hill. During the Jurassic, Strawberry Bank was within the Wessex Basin, which lay to the south and south east of the Severn Basin and encompassed most of southern England. Palaeogeographically, Strawberry Bank was similar to Alderton Hill, lying in the epicontinental seaway close to several small islands (Fig 1B). The sedimentary facies indicate that it is likely that Strawberry Bank was close to land [41]. The insect material originates from a single layer of pavement-like concretions of variable thickness, referred to as the Saurian and Fish Zone within the Barrington Member of the Beacon Limestone Formation [81, 89]. Similar to Alderton Hill, this

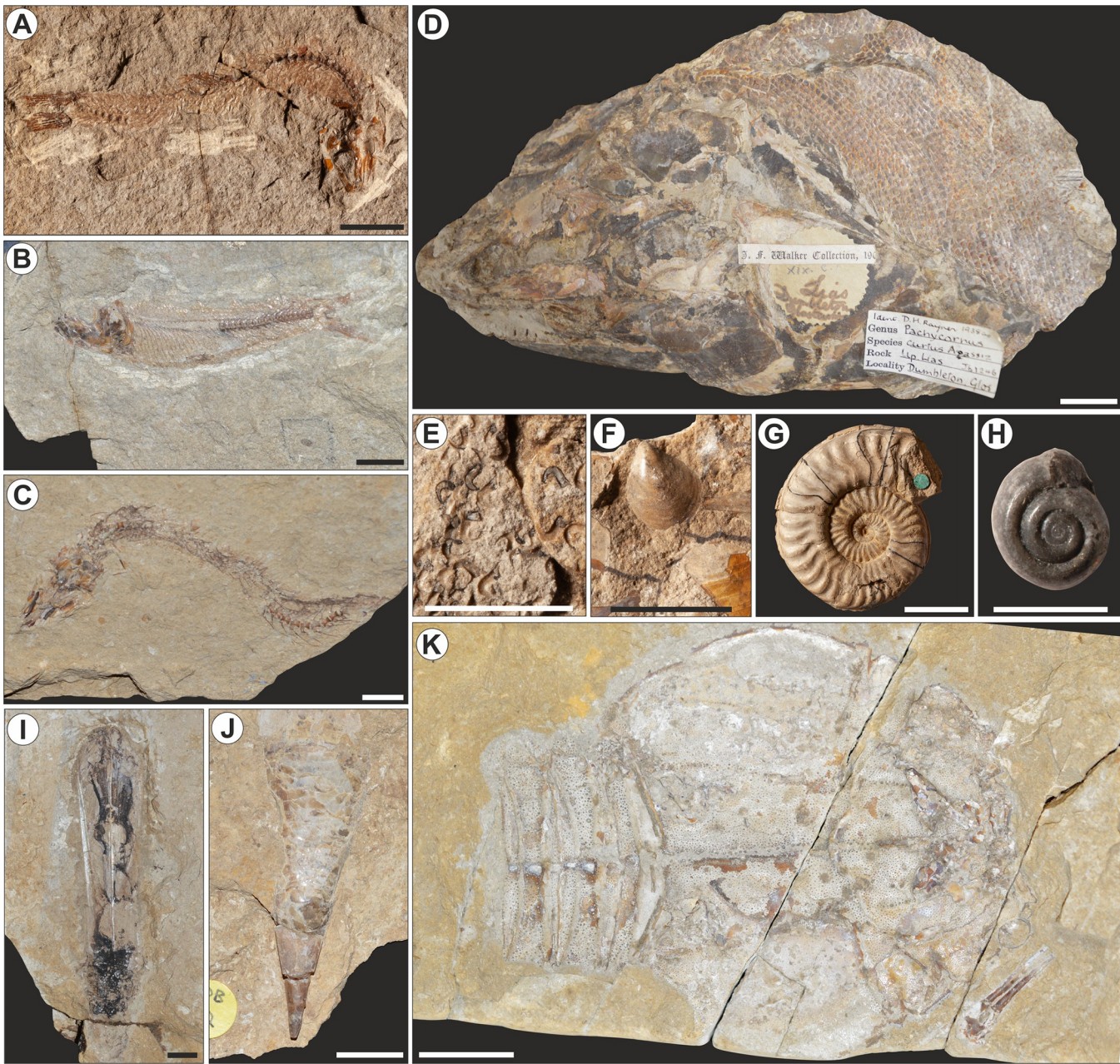

**Fig 2. Fauna associated with the insect horizon at Alderton Hill, Gloucestershire, UK.** Scale = 1 cm unless otherwise stated. **(A)** *Leptolepis concentricus* Egerton, 1849 [63], BGS 114078–79 (Actinopterygii: Leptolepiformes); **(B)** *L. normandica* Nybelin, 1962 [100], NHMUK PV P.7621, in association with indeterminate insect; **(C)** *L. coryphaenoides* Bronn, 1830 [101], NHMUK PV P.28855; **(D)** *Pachycormus macropterus* de Blainville, 1818 [102], CAMSM J.61246 (Pachycormiformes: Pachycormidae), scale = 2 cm; **(E)** Hooklets of *Geoteuthis* Münster, 1843 [103], BGS GSM 117962–63 (Cephalopoda), scale = 5 mm); **(F)** *Pseudomytiloides dubius* Sowerby, 1823 [104], BGS Geol. Soc. 4147 (Bivalvia: Myalinida); **(G)** *Hildaites murleyi* Moxon, 1841 [93], BGS GS.32040, genotype (Cephalopoda: Ammonitida); **(H)** *Coelodiscus minutus* Schübler, 1833 [105], BGS GSM 114054 (Gastropoda: Coelodiscidae), scale = 2.5 mm; **(I)** *Chondroteuthis wunnenbergi* Bode, 1933 [106], NHMUK PI C.5257a,b (Cephalopoda: Belemnitida); **(J)** *Chondroteuthis wunnenbergi* Bode, 1933 [106], NHMUK PI C.59301 (Cephalopoda: Belemnitida); **(K)** *Proeryon richardsoni* Woodward, 1911 [98], CHAGM F.685, holotype (Decapoda: Eryonidae), scale = 2 cm.

horizon occurs within the *exaratum* Subzone of the *falciferum* Zone and therefore records the T-OAE [41]. The concretions from the Strawberry Bank succession are thought to be part of a localised deposit, as they are not observed within the Toarcian succession near to the village of

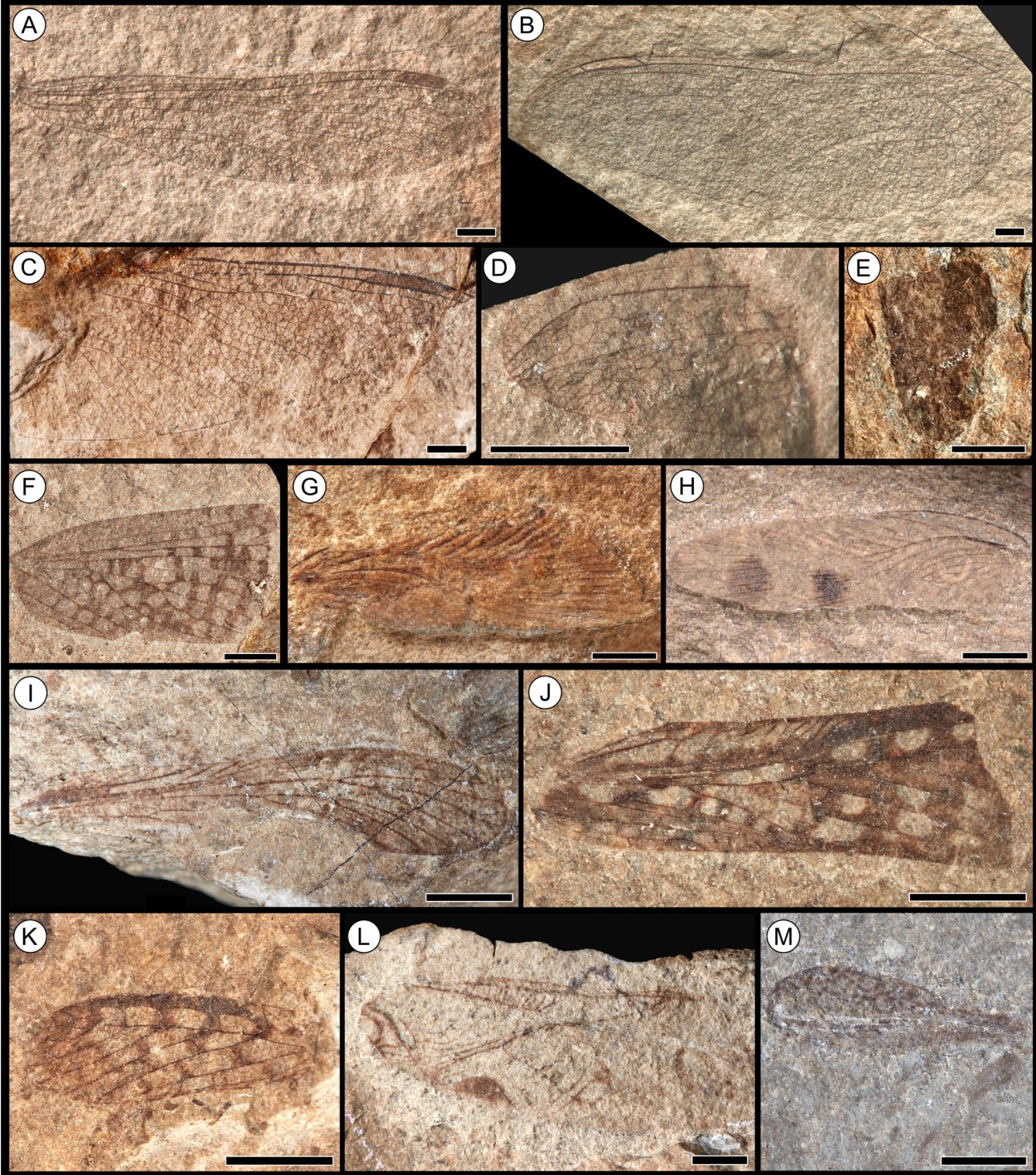

**Fig 3. Fossil insect orders Odonata,? Dermaptera, Reculida, Blattodea and Orthoptera from the Toarcian of Alderton Hill, Gloucestershire.** (A) *Heterophlebia buckmani* Brodie, 1845 [62] (Odonata: Heterophlebiidae), NHMUK PI I.11310; (B) *Heterothemis brodiei* Buckman, 1843 [60] (Odonata: Liassogomphidae), NHMUK PI I.3552 (holotype); (C) *H. brodiei* Buckman, 1843 [60], NHMUK PI I.11288; (D) *Protomyrmeleon* sp. Geinitz, 1887 [112] (Odonata: Protomyrmeleontidae), NHMUK PI In.59065; (E)? Dermaptera De Geer, 1773 [113], NHMUK PI I.11403; (F) *Geinitzia carpentieri* Zeuner, 1937 [71] (Reculida: Geinitziidae), NHMUK PI In.36200 (holotype); (G) *Liadoblattina blakei* Scudder, 1886 [67] (Blattodea: Raphidiomimidae), NHMUK PI I.3574 (holotype); (H) Blattodea Latreille, 1810 [114], undescribed, BRSMG Cg2374; (I) *Locustopsis* sp. Handlirsch, 1906 [115] (Orthoptera: Locustopsidae), NHMUK

PI I.11370; **(J)** "*Elcana*" cf. *geinitzi* (Heer, 1880) [116] (Orthoptera: Elcanidae), NHMUK PI I.11295; **(K)** "*Elcana*" cf. *geinitzi* (Heer, 1880) [116], NHMUK PI I.3560; **(L)** *Protogryllus magnus* Zeuner, 1937 [71] (Orthoptera: Protogryllidae), NHMUK PI I.11324 (holotype); **(M)** Orthoptera Olivier, 1789 [117], *incertae sedis*, NHMUK PI I.11405 (leg). Scale bars = 2.5 mm.

Seavington St. Michael, approximately 4 km to the east of Ilminster [89, 107]. While much of the fauna from the Strawberry Bank is exceptionally well-preserved, including phosphatized soft tissue and three-dimensional preservation of marine nekton [89], giving the location its Lagerstätte status, the insects are poorly preserved. This we attribute to the relatively coarse-grained nature of the layer that contains the insects (bioclastic wackestone to packstone with bioclasts of silt to coarse-sand size (100 to 700 μm)) [41].

## Materials and methods

Three hundred and sixty-three numbered rock specimens, that possess fossilised insects from Alderton Hill Quarry, Dumbleton Pit and Naunton Farm Quarry, were located in eight museum and institutional collections (Table 1). There are less specimens (n = 363) than individual insects (n = 370), as some specimens contained multiple individuals. The majority of the insect material from Alderton Hill was collected by Peter Bellinger Brodie and Thomas James Slatter and subsequently acquired by the Natural History Museum, London (NHMUK) in the late 19th century. Additional specimens from Alderton Hill located and used in this study were: those collected by Harriet Sophia Holland [64, 108] which are now in the collections of the Wilson Museum, Cheltenham (CHAGM); those at the British Geological Survey, Keyworth (BGS), National Museum of Wales, Cardiff (NMW), Warwickshire Museum, Warwick (WARMS), Sedgwick Museum, Cambridge (CAMSM) and Museum of Comparative Zoology, Harvard University (PALE); and most recently, nine insect specimens collected in the 1970s and 1980s by Mike J. Simms, which are now in the Bristol City Museum (BRSMG). A database was created of all the Alderton Hill insect specimens and the associated data, including the catalogue numbers, authors of the collection, taxonomy, type status, specimen descriptions and preservation for each fossil. For all material from NHMUK, specimen numbers are made up of a unique specimen number and a series of prefixes as follows: NHMUK–institution; PV–division (vertebrates); PI–division (invertebrates); I–section (insects); P–section (fish); C–section (cephalopods).

**Table 1. Quantity of numbered specimens and locations of all individual Toarcian insect collections from the three Alderton Hill exposures, Gloucestershire, UK, including the name and abbreviations for each museum and institution collection.**

| Museum / Institution | Number of specimens | | | Total |
|---|---|---|---|---|
| | Alderton Hill Quarry | Dumbleton Pit | Naunton Farm Quarry | |
| Natural History Museum, London, UK (NHMUK) | 87 | 202 | 0 | **289** |
| Wilson Museum, Cheltenham, UK (CHAGM) | 1 | 45 | 2 | **48** |
| Bristol Museum & Art Gallery, Bristol, UK (BRSMG) | 4 | 4 | 1 | **9** |
| British Geological Survey, Keyworth, UK (BGS) | 0 | 6 | 0 | **6** |
| Sedgwick Museum, Cambridge, UK (CAMSM) | 5 | 0 | 0 | **5** |
| Warwickshire Museum, Warwick, UK (WARMS) | 3 | 1 | 0 | **4** |
| Museum of Comparative Zoology, Harvard University, USA (PALE)[a] | 1 | 0 | 0 | **1** |
| National Museum Wales, Cardiff, UK (NMW) | 1 | 0 | 0 | **1** |
| **Total** | **102** | **258** | **3** | **363** |

[a] In addition to this specimen (PALE-8749), this collection also includes the counterpart (PALE-8751) to the holotype of *Paractinophlebia curtisii* (NHMUK PI I.3585).

**Table 2. Quantity of numbered specimens and locations of all individual Toarcian insect collections from Strawberry Bank, Ilminster, Somerset, UK, including the abbreviations for each museum and institution collection.**

| Museum / Institution | Number of specimens from Strawberry Bank |
|---|---|
| Somerset Heritage Centre, Taunton, UK (TTNCM) | 359 |
| Natural History Museum, London, UK (NHMUK) | 18 |
| Sedgwick Museum, Cambridge, UK (CAMSM) | 8 |
| Bristol Museum & Art Gallery, Bristol, UK (BRSMG) | 1 |
| **Total** | **386** |

Three hundred and eighty-six numbered rock specimens, that possessed fossilised insects from Strawberry Bank, were located within four museum and institutional collections (Table 2). Three hundred and fifty-nine insect specimens were examined within the collections of TTNCM; this material was originally collated and studied by Andrew J. Ross (National Museums Scotland), a list of preliminary identifications was supplied to the senior author of Williams *et al.* [41] in which the fauna was summarised. Additional specimens (n = 27) from Strawberry Bank were also discovered during this study within the collections of NHMUK (n = 18), CAMSM (n = 8) and BRSMG (n = 1). Again, there are less specimens (n = 386) than individual insects (n = 799). All of the insect material from Strawberry Bank, including the associated taxonomic and taphonomic data, was also compiled into a database for comparison with the Alderton Hill palaeoentomofauna. The four Toarcian insect specimens from Alderton Hill, documented as being held at Warwickshire Museum [109] could not be re-examined, as the collections were inaccessible at the time of this study. An additional six specimens from the collections at NHMUK and one specimen at Somerset Heritage Centre, Taunton, UK (TTNCM) were not re-examined due to either being lost (one specimen: NHMUK PI I.3988) or on loan. Taxonomic data for these specimens is therefore based on either Kelly [109] or the original museum databases.

## Taxonomic identification methods

Prior to this study, most of the Alderton Hill insects within the museum collections had been given a preliminary identification on a label stored with the specimen (largely by Edmund Aleksander Jarzembowski in the collections of NHMUK and CHAGM), but less than 15% of the specimens have previously been taxonomically described in the literature [see 66–72]. Within this study, through direct observation and high-resolution photographs, the taxonomy of each specimen was reviewed. This review included an update of the taxonomic nomenclature, confirmation of preliminary identifications, and where possible, refinement to a finer taxonomic resolution. The Strawberry Bank insect material, held within the collections of TTNCM and originally identified by Andrew J. Ross in Williams *et al.* [41] was also re-examined as part of this study, and in some cases, the taxonomy of these insects was refined.

For both the Alderton Hill and Strawberry Bank collections, every fragment that could be identified as insect-derived was recorded in order to reduce bias during identification. No additional specimen preparation was undertaken. Incomplete or poorly preserved material that could not confidently be assigned to a particular insect taxon was labelled 'indeterminate'. Following open nomenclature convention [110], where the taxonomic identification of a specimen is uncertain, a? is placed in front of the taxon name (e.g.? Dermaptera). Where possible, insects were examined with a binocular stereomicroscope at the different institutes to assess both the taxonomy and individual taphonomic grade. Following standard entomological nomenclature [111 and references therein], taxonomic identifications were predominantly based on wings and their venation.

Accurate morphological data and measurements of the preserved length and width of specimens were recorded with digital callipers to the nearest 0.1 mm. For articulated specimens, length was measured from the anterior to the posterior preserved margin of the insect, whilst width was measured across the widest section. For isolated wings, length was measured from wing tip to base, whilst width was measured across the widest dimension. However, minor vertical compression of the material during fossilization may have resulted in limited distortion, meaning that they don't accurately reflect those in life. The orientation (dorsoventral or lateral position) of each insect was also recorded. The feeding habits of the Alderton Hill insects were determined by direct comparison with present-day taxa [e.g. 111] where possible, whereas the feeding habits of the Strawberry Bank insects could not be inferred as most specimens could not be identified to sufficient taxonomic level.

As the collections at NHMUK contained the majority of the insect material, including many well-preserved specimens, all of the Alderton Hill and Strawberry Bank insects were photographed using a Canon EOS 5DS R with a single head light source with an adjacent reflector in order to best capture the preserved wing venation. Within the smaller collections e.g. CHAGM, BRSMG and BGS, the insect material was often poorly preserved and fragmentary, and therefore only well-preserved specimens were photographed. The Strawberry Bank insects at TTNCM were not photographed due to the poor and fragmentary preservation of the majority of the material; the best preserved examples are figured in Williams *et al.* [41]. Detailed photographs were also taken of associated fauna from Alderton Hill within the collections of NHMUK and CAMSM (camera model: Nikon D3200 with Nikon Micro-Nikkor 40mm F/2.8 macro lens). A specimen from each insect order, family, genus and species from Alderton Hill has been incorporated into palaeontological plates using CorelDRAW Graphics Suite X8 (Figs 3–5). In each case, we have chosen the best preserved specimen within each taxa. No plates are provided for the new Strawberry Bank specimens, due to the poor preservation of the material.

### Taphonomic classification method

The completeness and articulation of each insect specimen from Alderton Hill and Strawberry Bank was individually assessed using a standard semiquantitative method of grading the preservational quality (modified from [120]; Table 3). An alphanumeric code reflecting preservational quality was assigned to each individual based on the preserved morphological features, including the presence/absence of wings and appendages and their completeness, as explained below.

**Table 3. Description of the basis for designation of insect material into the primary and secondary taphonomic grades.** Modified from [120].

| Level of articulation | Primary taphonomic grades | | Secondary taphonomic grades | |
|---|---|---|---|---|
| | Grade | Description | Grade | Description |
| **Articulated material** | 1 | Body with forewings and/or hindwings | 1.1 | Complete body; with appendages |
| | | | 1.2 | Complete body; without appendages |
| | | | 1.3 | Body deformed; head, thorax, or abdomen lost / indistinguishable |
| **Disarticulated material** | 2a | Isolated body | 2a.1 | Almost complete body (≥75% complete) |
| | | | 2a.2 | Skeletal body element(s) e.g. isolated body fragments, leg etc. |
| | 2b | Isolated forewing/s | 2b.1 | Almost complete forewing/s (≥75% complete) |
| | | | 2b.2 | Forewing fragment/s |
| | 2c | Isolated hindwing/s | 2c.1 | Almost complete hindwing/s (≥75% complete) |
| | | | 2c.2 | Hindwing fragment/s |
| | 2d | Isolated unassigned wing/s | 2d.1 | Almost complete wing/s (≥75% complete) |
| | | | 2d.2 | Wing fragment/s |
| **Indeterminable** | ?? | Unidentifiable insect fragments | ?? | Unidentifiable insect fragments |

The specimens were first divided into three groups based on the degree of disarticulation and indeterminability (Table 3). The first group (1), including all specimens that possessed a body with forewings and/or hindwings, was then further subdivided into three secondary taphonomic grades based on the degree of completeness; grades ranged from 1.1 through 1.3, meaning that specimens assigned to 1.1 were of the highest quality of preservation and 1.3 the poorest. The second group (2) included four subgroups: isolated bodies (2a), forewing/s (2b), hindwings (2c) and unassigned wings (2d). Each subgroup was then further divided into two taphonomic grades, designated .1 and .2, based on the degree of completeness of each insect element (Table 3). Unidentifiable insect fragments could not be used for further analysis and therefore were assigned to the third group (labelled '??'). Insect orders that only possessed one pair of wings e.g. Diptera were included within 2b. As many specimens possessed a part and counterpart which often displayed varying taphonomic grades, the taphonomic assessment was based on the better-preserved one. To prevent researcher bias, by which an insect could be considered better preserved because it had been previously assigned to a taxon, any associated taxonomic information was not examined until after taphonomic grades had been assigned to the specimens.

## Statistical analysis

As taphonomic grades can be influenced by the anatomy of certain insects creating a preservational bias, a Chi-squared statistical test was performed in PAST 4.03 [121] to quantify and compare the association between each of the taphonomic grades and order diversity. In order to determine the degree to which each primary taphonomic grade contributed to the total Chi-squared score at order level, Pearson residuals were calculated for each entry in the Chi-squared contingency table at the order level using the statistics programme R (version 4.2.0). In order to determine how representative the fossil insect assemblages are at all taxonomic ranks at Alderton Hill and Strawberry Bank, individual-based rarefaction analyses were carried out at order-, family-, genus- and species-level using PAST 4.03 [121]. This allowed taxonomic richness at different sample sizes to be meaningfully compared [122] and assessment of whether sampling was sufficient to characterise the fossil community at each locality and taxonomic level. Standard errors (square roots of resampling variances) are converted to 95 percent confidence intervals in PAST 4.03 [121].

# Catalogue of identified taxa

The below catalogues for both entomofaunas are split into the most basal lineages (Ephemeroptera, Odonata, Dermaptera, Reculida, Blattodea, Orthoptera, and Hemiptera) and the holometabolous insects (Coleoptera, Necrotrichoptera, Mecoptera, Neuroptera, and Diptera), and follow standard entomological convention of the insect orders within these [e.g. 45]). Where there is no reference list the authority reference has been used.

*Institutional abbreviations*: **BGS,** British Geological Survey, Keyworth, UK; **BRSMG,** Bristol Museum & Art Gallery, Bristol, UK; **CAMSM,** Sedgwick Museum, Cambridge, UK; **CHAGM,** Wilson Museum, Cheltenham, UK; **NHMUK,** Natural History Museum, London, UK; **NMW,** National Museum Wales, Cardiff, UK; **PALE,** Museum of Comparative Zoology, Harvard University, USA; **TTNCM,** Somerset Heritage Centre, Taunton, UK; **WARMS,** Warwickshire Museum, Warwick, UK.

## Alderton Hill palaeoentomofauna

**Class** Insecta Linnaeus, 1758 [118]
 **Subclass** Pterygota Brauer, 1885 [123]

**Order** Ephemeroptera Hyatt & Arms, 1890 [124]

*Assigned material.*–WARMS G 8084.

*Remarks.*–Specimen was not available for re-examination (see materials and methods). WARMS G 8084 was originally reported as "*Ephemera*" by Kelly [109], however this determination is uncertain, as only two undescribed ephemeropteran wings are known from the Toarcian insect-bearing localities of Grimmen and Dobbertin, Germany.

**Order** Odonata Fabricius, 1793 [125]

**Suborder** Anisozygoptera Handlirsch, 1906 [115]

**Family** Heterophlebiidae Needham, 1903 [126]

**Genus** *Heterophlebia* Westwood, 1849 [127]

*Heterophlebia buckmani* (Brodie, 1845) [62]

(Fig 3A)

*Holotype.*–NHMUK PI I.11343, isolated partial forewing (33.8 mm long, 8.4 mm wide).

*Paratype.*–NHMUK PI I.3988; holotype of *Heterophlebia dislocata* Westwood, 1849 [127] (missing from collections, unable to be re-examined).

*Assigned material.*–NHMUK PI I.11310 & NHMUK PI I.11344, holotype of *Heterophlebia tillyardi* Handlirsch, 1939 [128]; isolated forewing, part and counterpart (33.8 mm long, 7.3 mm wide); BRSMG Cg2370 & BRSMG Cg2371, isolated partial forewing, part and counterpart (26 mm long, 8.4 mm wide); NHMUK PI I.59404 & NHMUK PI I.10496, isolated left hindwing, part and counterpart (35.8 mm long, 8.6 mm wide); NHMUK PI I.11312, holotype of *Heterophlebia angulata* Tillyard, 1925 [69]; isolated left partial male hindwing (20.7 mm long, 7.7 mm wide); NHMUK PI I.3317, two wing fragments (unable to be measured).

*Remarks.*–All additional material is fragmentary and the venation in these wing fragments is identical to those of other specimens attributed to *Heterophlebia buckmani* [see 73]. See [129] for an extensive synonymy list for *H. buckmani*.

*References.*–[69, 73, 115, 128, 129].

*Heterophlebia* sp.

*Assigned material.*–CHAGM F.651, partial wing fragment (12.1 mm long, 8.1 mm wide); CHAGM F.654, wing tip fragment (12.3 mm long, 8.8 mm wide).

**Family** Liassogomphidae Tillyard, 1935 [130]

**Genus** *Heterothemis* Handlirsch, 1906 [115]

*Heterothemis brodiei* (Buckman, 1843) [60]

(Fig 3B–3C)

*Holotype.*–NHMUK PI I.3552, isolated left female hindwing, lectotype selected as holotype (45 mm long, 13.9 mm wide).

*Assigned material.*–BGS Geol.Soc.Coll. 4126, isolated forewing apex (19.6 mm long, 3.4 mm wide); NHMUK PI I.11288 & NHMUK PI I.11269, isolated partial right hindwing, part and counterpart (31.5 mm long, 11 mm wide); CHAGM F.653, isolated partial basal hindwing (21.3 mm long, 12.6 mm wide); CHAGM F.657.1 & CHAGM F.657.2, partial wing fragment, part and counterpart (27.8 mm long, 8.7 mm wide); WARMS G 8079 (unable to be measured).

*Remarks.*–WARMS G 8079 was not available for examination but was reported as *Heterothemis brodiei* by Kelly [109]. All additional material is fragmentary and the venation in these wing fragments is identical to those of other specimens attributed to *Heterothemis brodiei* [see 73].

*References.*–[58, 73, 115, 128, 131].

*Heterothemis* sp.

*Assigned material.*–CHAGM F.650, partial wing fragment (13.3 mm long, 11.7 mm wide); CHAGM F.655, basal wing fragment (20.9 mm long, 5 mm wide); CHAGM F.662, partial wing tip (13.6 mm long, 4.6 mm wide); PALE-8749, partial hindwing fragment (unable to be measured).

*Remarks.*–PALE-8749 was originally reported as? *Liassogomphus* in the database of the Zoological Collections, Museum of Comparative Zoology, Harvard University.

**Suborder** Archizygoptera Handlirsch, 1906 [115]

**Family** Protomyrmeleontidae Handlirsch, 1906 [115]

**Genus** *Protomyrmeleon* Geinitz, 1887 [112]

*Protomyrmeleon* sp.

(Fig 3D)

*Assigned material.*–NHMUK PI In.59065, isolated partial supposed hindwing (6.8 mm long, 2.5 mm width).

*Remarks.*–NHMUK PI In.59065 possess venation identical to those of other specimens attributed to *Protomyrmeleon* [see 73].

*References.*–[132, 133].

**Family**? Protomyrmeleontidae Handlirsch, 1906 [115]

gen. et sp. indet

*Assigned material.*–NHMUK PI I.11350, isolated partial wing (5.7 mm length, 2 mm width).

**Family** *incertae sedis*

gen. et sp. indet

*Assigned material.*–Isolated partial forewing: BRSMG Cg2359. Isolated partial wings: NHMUK PI I.3330; NHMUK PI In.59078 & NHMUK PI In.59080, part and counterpart; CHAGM F.630; CHAGM F.652; CHAGM F.681; NHMUK PI In.59075; NHMUK PI I.11347 & NHMUK PI I.11375 (part and counterpart); CAMSM TN 3193.

*Remarks.*–Attributed to Odonata based on the dense cellular wing venation, but further taxonomic determination to family or genus level cannot be made due to fragmentary preservation.

**Order**? Dermaptera De Geer, 1773 [113]

**Family** *incertae sedis*

(Fig 3E)

*Assigned material.*–NHMUK PI I.11403, isolated elytron (5.6 mm long, 3.5 mm wide).

*Remarks.*–Only tentatively assigned to this order due to poor preservation of the specimen.

**Order** Reculida Handlirsch, 1906 [115]

**Family** Geinitziidae Handlirsch, 1906 [115]

**Genus** *Geinitzia* Handlirsch, 1906 [115]

*Geinitzia carpentieri* Zeuner, 1937 [71]

(Fig 3F)

*Holotype.*–NHMUK PI In.36200, isolated partial forewing, part and counterpart (18.4 mm long, 7.2 mm wide).

*Assigned material.*–BRSMG Cg2364 & BRSMG Cg2365, isolated partial forewing, part and counterpart (9.8 mm length and 7.8 mm width).

*Remarks.*–Additional material attributed to *Geinitzia carpentieri* based on species characteristics described by Zeuner [71], including: $MA_1$ with terminal fork; $CuA_2$ not undulated.

*References.*–[45, 47, 71, 115, 128].

**Superorder** Dictyoptera Latreille, 1829 [134]

**Order** Blattodea Latreille, 1810 [114]

**Family**? Raphidiomimidae Vishnyakova, 1973 [135]

**Genus** *Liadoblattina* Handlirsch, 1906 [115]

*Liadoblattina blakei* (Scudder, 1886) [67]

(Fig 3G)

*Holotype.*–NHMUK PI I.3574, isolated tegmen (14.6 mm long, 5 mm wide).

*Assigned material.*–NHMUK PI I.11361, tegmen and pronotum (2.5 mm long, 1.5 mm wide); NHMUK PI I.3562, holotype of *Mesoblattina bensoni* Scudder, 1886 [67], isolated tegmen (18 mm long, 5 mm wide); NHMUK PI I.11341, isolated partial tegmen, part and counterpart (16.1 mm long, 5 mm wide).

*Remarks.*–Attributed to *Liadoblattina blakei* based on species characteristics described by Vršanský and Ansorge [75], including an elongate tegmen, which is up to three times as long (*c.* 20 mm) as it is wide, and a clavus that is more than twice as long as it is wide.

*References.*–[67, 75, 115, 128].

**Family** uncertain

undescribed(Fig 3H)

*Assigned material.*–BRSMG Cg2374 & BRSMG Cg2375, isolated tegmen, part and counterpart (12.8 mm long, 3.8 mm wide).

*Remarks.*–Our observations show that this specimen is noticeably dissimilar to other known Toarcian cockroaches and represents a new genus and species. The family placement of this specimen is uncertain at present.

**Family** *incertae sedis*

gen. et sp. indet

*Assigned material.*–CHAGM F.660, isolated partial wing (6.8 mm long, 2 mm wide).

*Remarks.*–Attributed to Blattodea based on the tegmen venation, which involves numerous closely spaced parallel veins, however further taxonomic determination to family or genus level cannot be made due to poor venation preservation.

**Order** Orthoptera Olivier, 1789 [117]

**Family** Locustopsidae Handlirsch, 1906 [115]

**Genus** *Locustopsis* Handlirsch, 1906 [115]

*Locustopsis* sp.

(Fig 3I)

*Assigned material.*–Isolated forewings: NHMUK PI I.11303; NHMUK PI I.11370. Isolated partial forewings: NHMUK PI I.11334; NHMUK PI I.11330; NHMUK PI I.11363; NHMUK PI I.11351. Isolated hindwing: NHMUK PI I.11340.

*Remarks.*–Specimens possess venation and colour patterns similar to those of other specimens attributed to the genus *Locustopsis* [see 136].

*References.*–[115, 128, 136, 137].

**Family** Locustopsidae Handlirsch, 1906 [115]

gen. et sp. indet

*Assigned material.*–BGS GSM 117360, isolated partial wing (8.7 mm long, 2.5 mm wide).

**Family** Elcanidae Handlirsch, 1906 [115]

**Genus** "*Elcana*" Giebel, 1856 [138]

*Remarks.*–Gorochov *et al.* [139] synonymized *Elcana* Giebel, 1856 [138] with *Panorpidium* Westwood, 1854 [65] as both generic names were based on the same specimen, the type species of *P. tesselatum* Westwood, 1854 [65]. Gorochov *et al.* [139] also divided Elcanidae in two subfamilies: Elcaninae Handlirsch 1906 [115] [= Baisselcaninae Gorochov, 1986 [140]] and Archelcaninae Gorochov, Jarzembowski & Coram, 2006 [139].

The numerous species of *Elcana* described by earlier authors [see 115, 141, 142] are in need of taxonomic revision [139]. Although a possible generic name for Early Jurassic elcanids is *Clathrotermes* Heer, 1865 [143], here we use the name "*Elcana*" as a placeholder until a full revision of this genus is undertaken. At species level, we follow Zessin [142] in his treatment of Early Toarcian elcanid species, who reduced more than 100 Lower Toarcian species described by Handlirsch [115, 128] and Bode [45, 47] from Dobbertin and the vicinity of Braunschweig to four: "*Elcana*" *minima* Handlirsch, 1906 [115], "*Elcana*" *geinitzi* (Heer, 1880) [116],

"*Elcana*" *media* Handlirsch, 1906 [115] and "*Elcana*" *magna* Handlirsch, 1906 [115]. Recently, Heads *et al.* [144] described two new elcanid species from the Lower Toarcian of Luxembourg as *Archelcana tina* Heads *et al.*, 2022 [144] and *A. numbergerae* Heads *et al.*, 2022 [144]. However, these species may prove to be junior synonyms.

"*Elcana*" cf. *geinitzi* (Heer, 1880) [116]

(Fig 3J–3K)

**Holotype.**–MBI 15.1 a,b, isolated forewing. Grimmen Formation, 'Green Series'; Early Jurassic: Lower Toarcian of Dobbertin, Mecklenburg-Vorpommern, Germany.

**Assigned material.**–NHMUK PI I.11427, isolated partial forewing (8.1 mm long, 2.1 mm wide); NHMUK PI I.11295, isolated partial forewing (10 mm long, 2.8 mm wide); NHMUK PI I.3349 (specimen was not available to be measured); NHMUK PI I.3560, holotype of *Elcana brodiei* (Handlirsch, 1906) [115], isolated partial forewing (7.2 mm long, 2.6 mm wide).

**Remarks.**–NHMUK PI I.3349 was reported as *Archelcana geinitzi* by Kelly [109].

**References.**–[46, 115, 116, 128, 142, 145].

"*Elcana*" sp.

**Assigned material.**–NHMUK PI I.11394, isolated partial forewing tip (5.1 mm long, 2 mm wide); NHMUK PI I.38, part and counterpart (9.2 mm long, 4.5 mm wide)

**Family** Protogryllidae Zeuner, 1937 [71]

**Genus** *Protogryllus* Handlirsch, 1906 [115]

*Protogryllus* cf. *dobbertinensis* (Geinitz,1880) [116]

**Holotype.**–FGWG 117/7, isolated forewing. Grimmen Formation, 'Green Series'; Early Jurassic: Lower Toarcian of Dobbertin, Mecklenburg-Vorpommern, Germany.

**Assigned material.**–NHMUK PI I.11421, isolated partial male forewing (16.4 mm long, 5.3 mm wide).

**References.**–[44, 47, 71, 115, 128].

*Protogryllus magnus* Zeuner, 1937 [71]

(Fig 3L)

**Holotype.**–NHMUK PI I.11324, isolated partial male forewing (27 mm long, 5.7 mm wide).

**References.**–[71].

*Protogryllus* sp.

**Assigned material.**–NHMUK PI I.11377, partial body with wings (17.7 mm long, 2.7 mm wide); NHMUK PI I.11461, isolated partial wing (12.3 mm long and 3.7 wide).

**Family** *incertae sedis*

gen. et sp. indet

(Fig 3M)

**Assigned material.**–Almost complete individuals: NHMUK PI I.11270. Isolated exoskeletal elements: NHMUK PI I.3314; NHMUK PI I.11321; NHMUK PI I.11405; NHMUK PI I.11333; NHMUK PI I.11392. Isolated partial wings: BGS GSM 117359; BGS GSM 117361; NHMUK PI I.3305; NHMUK PI I.11301; NHMUK PI I.11290; NHMUK PI I.11396; NHMUK PI I.3304; NHMUK PI I.11264. Indeterminable: NHMUK PI I.11342; NHMUK PI I.11332; NHMUK PI I.59084.

**Remarks.**–Attributed to Orthoptera based on the tegminised forewings and distinctive colouration patterns. Further determination to a lower taxonomic level cannot be made due to fragmentary wing preservation.

**Order** Hemiptera Linnaeus, 1758 [118]

**Suborder** Fulgoromorpha Evans, 1946 [146]

**Family** Fulgoridiidae Handlirsch, 1939 [128]

**Genus** *Fulgoridium* Handlirsch, 1906 [115]

*Fulgoridium* sp.

(Fig 4A)

*Assigned material.*–Isolated skeletal elements: NHMUK PI I.11424. Isolated forewings: NHMUK PI I.3331 & NHMUK PI I.3334, part and counterpart; NHMUK PI I.11328. Isolated partial forewing: NHMUK PI I.3079. Isolated partial wings: NHMUK PI I.11275; NHMUK PI I.11293; NHMUK PI I.11315. Indeterminable: NHMUK PI I.11278; NHMUK PI I.11453.

*Remarks.*–Specimens NHMUK PI I.11278 and NHMUK PI I.11453 were not available for re-examination but were reported as *Fulgoridium* by Kelly [109]. From the Toarcian, only the genera *Fulgoridulum* Handlirsch, 1939 [128], *Margaroptilon* Handlirsch, 1906 [115], and *Elasmoscelidium* Martynov, 1926 [147] are currently well defined. Until a revision of all species of *Fulgoridium* is undertaken, it is impossible to determine the taxonomy of the Alderton Hill specimens more precisely.

*References.*–[44, 47, 115, 128, 147].

**Genus** *Margaroptilon* Handlirsch, 1906 [115]

*Margaroptilon brodiei* Handlirsch, 1906 [115]

(Fig 4B, 4C)

1906 *Margaroptilon brodiei* Handlirsch: 499

1906 *Margaroptilon woodwardi* Handlirsch: 499 syn. nov.

1906 *Margaroptilon bulleni* Handlirsch: 499 syn. nov.

*Holotype.*–NHMUK PI I.3561, spotted isolated partial forewing (6 mm long, 2.1 mm wide); NHMUK PI I.11266 (holotype of *M. bulleni*) & NHMUK PI I.11286 (holotype of *M. woodwardi*), isolated forewing, part and counterpart (6.3 mm long, 2.5 mm wide).

*Assigned material.*–Isolated forewings: CHAGM F.642; CHAGM F.664; NHMUK PI I.11296. Isolated partial forewings: CHAGM F.659.1 & CHAGM F.659.2 (part and counterpart); CHAGM F.661; CHAGM F.635; NHMUK PI I.11374; NHMUK PI I.11399; NHMUK PI I.11414; NHMUK PI I.11426; NHMUK PI I.11429; NHMUK PI I.15015; NHMUK PI I.3341; NHMUK PI I.11268; NHMUK PI I.11415. Indeterminable: NHMUK PI I.3309.

*Remarks.*–Following the suggestion in Kelly's PhD thesis [109], here we have synonymized *Margaroptilon brodiei* Handlirsch, 1906 [115]; *Margaroptilon bulleni* Handlirsch (1906) [115]; and *Margaroptilon woodwardi* Handlirsch, 1908 [115] based on morphological similarity, with the latter being the senior synonym. He recognized that *H. woodwardi* and *M. bulleni* were based on part and counterpart of the same wing. We regard the small differences in spot arrangement between *M. woodwardi* and *M. brodiei* as intraspecific variation, as Kelly [109] did. However, the name *M. brodiei* has priority instead of *M. woodwardi*. NHMUK PI I.3309 was not available for re-examination but was reported as *Margaroptilon woodwardi* by Kelly [109]. *Margaroptilon* is a planthopper with faint venation typical for fulgoromorphans, although it is difficult to study as the well sclerotized wing spots resemble those of *Fulgoridulum*.

*References.*–[46, 109, 115].

**Family** Fulgoridiidae Handlirsch, 1939 [128]

gen. et sp. indet.

*Assigned material.*–CHAGM F.636, isolated partial wing tip (7 mm long, 3.1 mm wide); CAMSM TN 3195, partial spotted wing (5.6 mm long, 2.7 mm wide).

**Family** Archijassidae Becker-Migdisova, 1962 [148]

**Genus** *Archijassus* Handlirsch, 1906 [115]

*Archijassus heeri* (Geinitz, 1880) [116]

(Fig 4D)

*Assigned material.*–NHMUK PI I.11458, isolated forewing (specimen not available for re-examination).

*References.*–[44, 116].

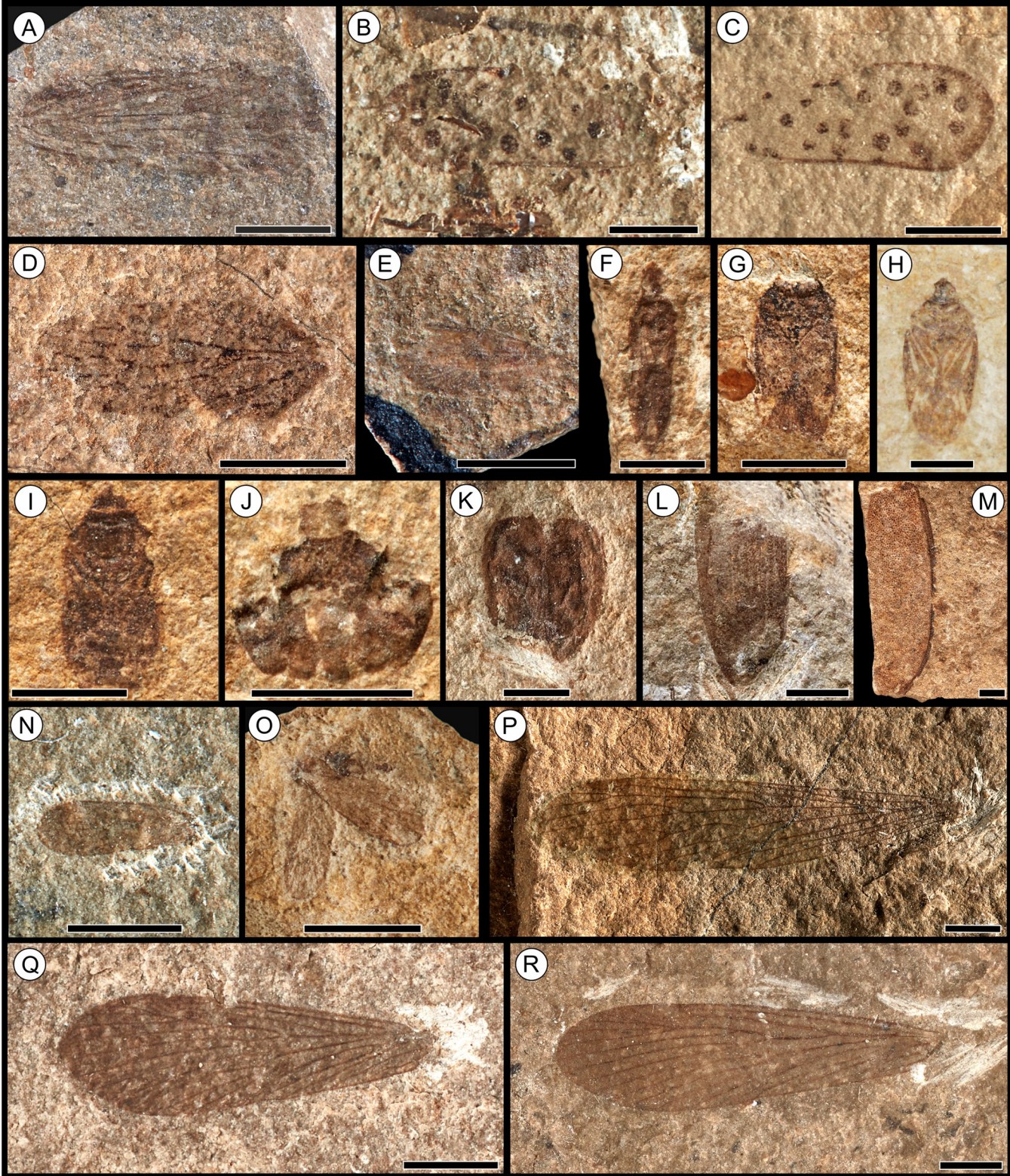

**Fig 4. Fossil insect orders Hemiptera, Coleoptera, Necrotrichoptera, Mecoptera from the Toarcian of Alderton Hill, Gloucestershire.** (A) *Fulgoridium* sp. Handlirsch, 1906 [115] (Hemiptera: Fulgoridiidae), NHMUK PI I.11328; (B) *Margaroptilon brodiei* Handlirsch, 1906 [115] (Hemiptera: Fulgoridiidae), NHMUK PI I.3561 (holotype); (C) *M. brodiei* Handlirsch, 1906 [115], NHMUK PI I.11266 (holotype of *M. bulleni*); (D) *Archijassus heeri* Geinitz, 1880 [116], (Hemiptera: Archijassidae), NHMUK PI I.11458; (E) Progonocimicidae Handlirsch, 1906 [115] (Hemiptera), NHMUK PI I.11465; (F) *Engynabis tenuis* Bode,

1953 [46] (Hemiptera), NHMUK PI I.3313; **(G)** Hemiptera Linnaeus 1758 [118], *incertae sedis*, NHMUK PI I.11446; **(H)** Hemiptera Linnaeus, 1758 [118], undetermined, CHAGM F.643.1; specimen photographed and reproduced with permission; **(I)** Hemiptera Linnaeus, 1758 [118], *incertae sedis*, NHMUK PI I.11317; **(J)** Coleoptera Linnaeus, 1758 [118], *incertae sedis*, NHMUK PI I.69; **(K)** Coleoptera Linnaeus, 1758 [118], *incertae sedis*, NHMUK PI I.11345; **(L)** Coleoptera Linnaeus, 1758 [118], *incertae sedis*, NHMUK PI I.3077; **(M)** Coleoptera Linnaeus, 1758 [118], *incertae sedis*, NHMUK PI I.11355; **(N)** *Necrotaulius parvulus* (Geinitz, 1884) [66] (Necrotrichoptera: Necrotauliidae), NHMUK PI I.15014; **(O)** *N. parvulus* (Geinitz, 1884) [66], NHMUK PI I.11389; **(P)** *Protobittacus handlirschi* Tillyard, 1933 [70] (Mecoptera: Bittacidae), composite photograph of the part (NHMUK PI I.11406) and counterpart (NHMUK PI I.11335) (holotype); **(Q)** *Orthophlebia brodiei* Tillyard, 1933 [70] (Mecoptera: Orthophlebiidae), NHMUK PI I.15019 (holotype); **(R)** *Orthophlebia* sp. Westwood, 1845 [62] NHMUK PI I.11225. Scale bars = 2.5 mm.

**Suborder** Coleorrhyncha Myers and China, 1929 [149]

**Family** Progonocimicidae Handlirsch, 1906 [115]

gen. et sp. indet.

(Fig 4E)

***Assigned material.***–NHMUK PI I.11274, isolated partial body (3.3 mm long, 1.1 mm wide); NHMUK PI I.11465, isolated forewing (4 mm long).

**Suborder** Heteroptera Latreille, 1810 [114]

**Infraorder** Gerromorpha Popov, 1971 [150]

**Genus** *Engynabis* Bode, 1953 [46]

*Engynabis tenuis* Bode, 1953 [46]

(Fig 4F)

***Assigned material.***–NHMUK PI I.3313, elongated body with paired wings (5.3 mm long, 1.2 mm wide).

***References.***–[46, 151].

**Family** *incertae sedis*

gen. et sp. indet.

(Fig 4G–4I)

***Assigned material.***–Partial individuals: NHMUK PI I.11362; NHMUK PI I.11446; CHAGM F.643.1 & CHAGM F.643.2. Isolated bodies: NHMUK PI I.11280; NHMUK PI I.11294; NHMUK PI I.11297; NHMUK PI I.11317; NHMUK PI I.82 & NHMUK PI I.74 (part and counterpart); NHMUK PI I.11379. Isolated skeletal elements: NHMUK PI I.11454; NHMUK PI I.3052; NHMUK PI I.11285; NHMUK PI I.11456; NHMUK PI I.11452; NHMUK PI I.3324; NHMUK PI I.68; NHMUK PI I.11445; BGS GSM 117358; CHAGM F.634.1 & CHAGM F.634.2 (part and counterpart). Isolated forewing: NHMUK PI I.78. Isolated partial hindwings: NHMUK PI I.11439. Isolated partial wings: NHMUK PI I.3342; CHAGM F.671 & CHAGM F.672 (part and counterpart); CHAGM F.670.1 & CHAGM F.670.2 (part and counterpart). Indeterminate: NHMUK PI In. 59079.

**Order** Coleoptera Linnaeus, 1758 [118]

gen. et sp. indet.

(Fig 4J–4M)

***Assigned material.***–Almost complete individuals: CHAGM F.645; CHAGM F.646; NHMUK PI I.11339; NHMUK PI I.11360. Partial individuals: CHAGM F.638; CHAGM F.647.1 & CHAGM F.647.2; CHAGM F.649.1 & CHAGM F.649.2; NHMUK PI I.3307; NHMUK PI I.11345; NHMUK PI I.11383; CAMSM TN 3196 a+b. Isolated bodies: NHMUK PI I.11267; NHMUK PI I.11287; NHMUK PI I.11292. Isolated skeletal elements: NHMUK PI I.69; NHMUK PI I.3308; NHMUK PI I.11302; NHMUK PI I.11404; NHMUK PI I.81. Isolated forewings: CHAGM F.640; CHAGM F.644; CHAGM F.663; CHAGM F.667; NHMUK PI I.3052; NHMUK PI I.3077; NHMUK PI I.3078; NHMUK PI I.11318; NHMUK PI I.11331; NHMUK PI I.11353 & NHMUK PI I.11354; NHMUK PI I.11355 & NHMUK PI I.11376; NHMUK PI I.11359; NHMUK PI I.11369; NHMUK PI I.11411; NHMUK PI I.11420;

NHMUK PI I.11431; NHMUK PI I.11432; NHMUK PI I.11438; NHMUK PI I.11441; NHMUK PI I.59071. Isolated partial forewings: CHAGM F.639; CHAGM F.641; NHMUK PI I.3335; NHMUK PI I.11368. Indeterminable: NHMUK PI I.2163; NHMUK PI I.3339; NHMUK PI I.3348 & NHMUK PI I.11308 (part and counterpart); NHMUK PI In. 59072; NHMUK PI In.59286; NHMUK PI In.59285; NHMUK PI In.59287; NHMUK PI In.59289; NHMUK PI In.59294; NHMUK PI I.11381.

*Remarks.*–Attributed to Coleoptera based on the forewings which are modified into highly sclerotised elytra. However, the majority of the Coleoptera are represented by disarticulated and fragmented elytra which are compressed, poorly preserved, and therefore difficult to identify beyond order level. Therefore, all beetles specimens from Alderton Hill remain unassigned, and no types are based on the recorded specimens. However, at least three elytra morphotypes can be identified, including smooth (NHMUK PI I.11345; Fig 4K), striated (NHMUK PI I.3077; Fig 4L) and pitted (NHMUK PI I.11355 and NHMUK PI I.11376; Fig 4M), indicating that the beetle fauna is likely to consist of several genera or species. The thoracic fragment NHMUK PI I.3052 was originally assigned to the recent genus *Coccinella* Linnaeus, 1758 [118] based on the associated specimen label, however this identification cannot be supported.

**Order** Necrotrichoptera Engel, 2022 [152]
**Family** Necrotauliidae Handlirsch, 1906 [115]
**Genus** *Necrotaulius* Handlirsch, 1906 [115]
*Necrotaulius parvulus* (Geinitz, 1884) [66]
(Fig 4N –4O)

*Holotype.*–FGWG 119/7, isolated wing. Grimmen Formation, 'Green Series'; Early Jurassic: Lower Toarcian of Dobbertin, Mecklenburg-Vorpommern, Germany.

*Assigned material.*–NHMUK PI I.15016 & NHMUK PI I.15014 (part and counterpart) isolated forewing, holotype of *Necrotaulius pygmaeus* Tillyard, 1933 [70] (3.2 mm long, 0.9 mm wide); NHMUK PI I.11425, isolated forewing (2.8 mm long, 0.7 mm wide); NHMUK PI I.3336, isolated body (3.8 mm long, 1.8 mm wide); NHMUK PI I.11389 (specimen not available for re-examination).

*Remarks.*–*Necrotaulius parvulus* is distinct due to the dense coverage of the wing with hairs. Wing venation of this species is similar to Trichoptera, however it has no distally bent CuP, a synapomorphic character of Trichoptera [77, 153]. Because of this CuP character, we here exclude *Austaulius* Thomson, Ross and Coram, 2018 [43] from Necrotauliidae and transfer it to Trichoptera *incertae familiae*.

*References.*–[66, 77, 109, 112, 128, 153–156].

?*Necrotaulius* sp.

*Assigned material.*–CHAGM F.637, isolated partial wing (specimen unable to be measured).

**Order** Mecoptera Packard, 1886 [157]
**Family** Bittacidae Handlirsch, 1906 [115]
**Genus** *Protobittacus* Tillyard, 1933 [70]
*Protobittacus handlirschi* Tillyard, 1933 [70]
(Fig 4P)

*Holotype.*–NHMUK PI I.11406 & NHMUK PI I.11335, isolated hindwing, part and counterpart (19.4 mm long, 4.3 mm wide).

*Remarks*. During our investigation of the Alderton Hill insect material, we located the unknown counterpart of the holotype (NHMUK PI I.11335), which displays most of the proximal part of the wing including the unique type of A1/A2 fork (Fig 4P).

*References.*–[70].

Bittacidae gen. et sp. indet.

***Assigned material.***–CHAGM F.632, isolated wing (8.5 mm long, 2 mm wide).

**Family** Orthophlebiidae Handlirsch, 1906 [115]

**Genus** *Orthophlebia* Westwood, 1845 [62]

*Orthophlebia brodiei* Tillyard, 1933 [70]

(Fig 4Q)

***Holotype.***–NHMUK PI I.15019, isolated forewing (10.7 mm long, 3 mm wide).

***Remarks.***–Note that Tillyard 1933 [70] referred to the holotype specimen as NHMUK PI I.15017; see Tillyard [70] for the original holotype description.

***References.***–[44, 70, 160, 161].

*Orthophlebia* sp.

(Fig 4R)

***Assigned material.***–Isolated forewings: NHMUK PI I.11413; NHMUK PI I.11418 & NHMUK PI I.11395 (part and counterpart); NHMUK PI I.11225; NHMUK PI I.11387. Isolated partial forewing: NHMUK PI I.11271. Isolated hindwing: NHMUK PI I.15013. Isolated partial hindwing: NHMUK PI I.11366. Isolated wing: NHMUK PI I.11448. Isolated partial wing: CAMSM TN 3192.

***References.***–[44, 158, 159].

**Family** Pseudopolycentropodidae Handlirsch, 1920 [160]

**Genus** *Pseudopolycentropus* Handlirsch, 1906 [115]

*Pseudopolycentropus* sp.

***Assigned material.***–CHAGM F.656, isolated forewing (5.4 mm long, 2 mm wide).

***References.***–[44, 47, 115, 128, 161].

**Order** Neuroptera Linnaeus, 1758 [118]

**Family** Osmylopsychopidae Martynova, 1949 [158]

**Genus** *Actinophlebia* Handlirsch, 1906 [115]

*Actinophlebia intermixta* (Scudder, 1885) [119]

(Fig 5A, 5B)

***Holotype.***–NHMUK PI I.3577 & NHMUK PI I.11346, isolated forewing, part and counterpart (9.2 mm length, 5.5 mm width). *Actinophlebia anglicana* Tillyard, 1933 [70] (NHMUK PI I.11346) is a counterpart of *Actinophlebia intermixta* (Scudder, 1885) [119], an objective synonym of the latter.

***Assigned material.***–NHMUK PI I.11349, isolated forewing (6.3 mm length and 4.1 mm width); NHMUK PI I.11319, isolated partial forewing (7.3 mm length and 4.5 mm width).

***Remarks.***–Neuropterans from the Toarcian with a triangular wing shape are often attributed to the genus *Actinophlebia* [44, 68, 70, 115]. However, Ponomarenko [162] regarded similar wings with a distally fused Sc with RA as belonging to *Parhemerobius* Bode, 1953 [46]. In *Actinophlebia*, the Sc appears to run free into the costal margin. This, however, is most likely based on a misinterpretation of the course of Sc, as this featured can be observed in the *Actinophlebia* sp. specimen LGA 1080 from the Lower Toarcian of Grimmen, Germany [see 45]. Note that Tillyard [70] originally measured NHMUK PI I.11346 at 11.2 mmm long, while Scudder [119] measured a length of 10 mm.

***References.***–[44, 47, 68, 70, 115, 119, 128, 162–165].

*Actinophlebia* sp.

***Assigned material.***–NHMUK PI I.11314, isolated partial wing (6.8 mm length, 4.4 mm width).

"Familia nova A" Makarkin *et al.*, 2012 [166]

**Genus** *Paractinophlebia* Handlirsch, 1906 [115]

*Paractinophlebia curtisii* (Scudder, 1886) [67]

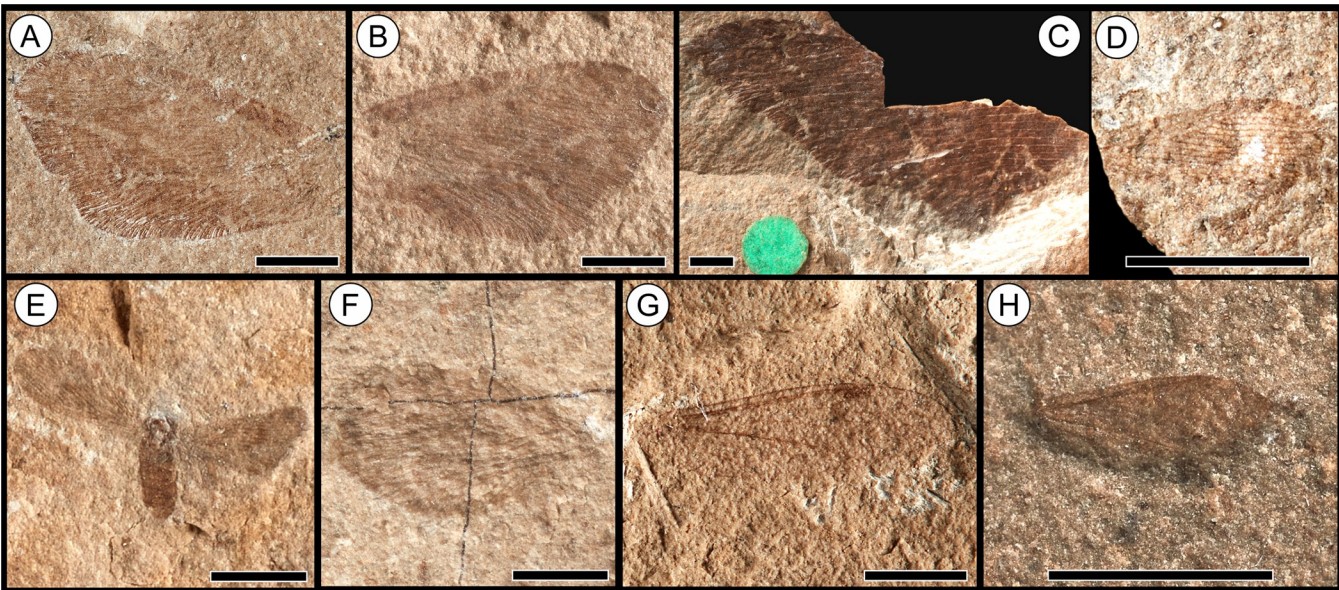

**Fig 5. Fossil insect orders neuroptera and diptera from the Toarcian of Alderton Hill, Gloucestershire. (A, B)** *Actinophlebia intermixta* Scudder, 1885 [119] (Neuroptera: Osmylopsychopidae), NHMUK PI I.3577 & NHMUK PI I.11346 (holotype, part and counterpart); **(C)** *Paractinophlebia curtisii* Scudder, 1886 [67] (Neuroptera: "Familia nova A"), NHMUK PI I.3585 (holotype); **(D)** *"Prohemerobius"aldertonensis* Whalley, 1988 [68] (Neuroptera: "Prohemerobiidae"), NHMUK PI I.11304 (holotype); **(E)** *"Archeosmylus" alysius* Whalley, 1988 [68] (Neuroptera: "Prohemerobiidae"), NHMUK PI I.3318 (holotype); **(F)** *"A." complexus* Whalley, 1988 [68] (Neuroptera: "Prohemerobiidae"), NHMUK PI I.11412 (holotype); **(G)** *Architipula anglicana* (Tillyard, 1933) [70] (Diptera: Limoniidae), NHMUK PI I.11298 (holotype); **(H)** *Grimmenia tillyardi* Kopeć *et al.*, 2017b [76] (Diptera: Limoniidae), NHMUK PI I.3328 (holotype). Scale bars = 2.5 mm.

(Fig 5C)

*Holotype.*–NHMUK PI I.3585, isolated partial hindwing (27.1 mm long, 7.9 mm wide); the counterpart (PALE-8751) is housed in the Entomology collection at the Museum of Comparative Zoology, Harvard University, and assigned as Neuroptera sp.

*Assigned material.*–NHMUK PI I.11320, isolated partial wing fragment (17.9 mm long, 9 mm wide).

*Remarks.*–Scudder [67] figured the counterpart (plate 48, Fig 16) at PALE. The holotype is most likely a hindwing fragment, which is strongly dissimilar to the wings of Kalligrammatidae [see 68], in particular by the presence of rare cross-veins arranged in series and by the very narrow subcostal space [167].

*References.*–[67, 166, 167].

**Family** *"Prohemerobiidae"*Handlirsch, 1906 [115]

**Genus** *"Prohemerobius"*Handlirsch, 1906 [115]

*"Prohemerobius"aldertonensis* Whalley, 1988 [68]

(Fig 5D)

*Holotype.*–NHMUK PI I.11304, isolated partial forewing (4 mm long, 1.5 mm wide).

*Assigned material.*–NHMUK PI I.11279, isolated partial wing (5.7 mm long, 2.7 mm wide); NHMUK PI I.11390, isolated partial wing (6.6 mm long, 1.8 mm wide).

*Remarks.*–The genus *Prohemerobius* Handlirsch 1906 [115] is regarded as a waste basket taxon for small Lower Jurassic neuropterans, and therefore requires a complete revision. Ponomarenko, 1995 [161] regards the English Lower Toarcian species *Archeosmylus alysius* Whalley, 1988 [68] and *A. complexus* Whalley, 1988 [68] as belonging to *Prohemerobius*. The only difference is whether the Sc and RA run parallel or strongly converge. We refer here the assignment of specimen to the certain taxa as provided by Whalley [68].

*References.*–[68, 162, 168].

*"Prohemerobius"* sp.

*Assigned material.*–NHMUK PI I.11284, isolated wing (7.4 mm length, 3.3 mm width); NHMUK PI Ig 60, isolated wing (9.1 mm length, 2.4 mm width); NHMUK PI I.11417, partial individual with wings (specimen was not available for measurement).

*"Archeosmylus" alysius* Whalley, 1988 [68]

(Fig 5E)

*Holotype.*–NHMUK PI I.3318, body (2.3 mm long, 0.5 mm wide) with attached wings (4.3 mm long, 1.7 mm wide), part and counterpart.

*Paratypes.*–NHMUK PI I.11416, isolated wing (5 mm long, 2.5 mm wide); NHMUK PI I.11433, partial individual with paired wings (3.5 mm long, 1.9 mm wide).

*Assigned material.*–NHMUK PI I.11281, isolated wing (3.7 mm long, 1.8 mm wide); NHMUK PI I.11402, isolated partial wing (6.5 mm long, 1.7 mm wide); NHMUK PI I.11347 & NHMUK PI I.11375, partial individual, part and counterpart (specimen could not be measured).

*References.*–[68, 162].

*"Archeosmylus" complexus* Whalley, 1988 [68]

(Fig 5F)

*Holotype.*–NHMUK PI I.11412, isolated wing (6.7 mm long, 3.7 mm wide).

*Paratype.*–NHMUK PI I.3315 & NHMUK PI I.3380, isolated wing, part and counterpart (9.1 mm, 2.1 mm wide).

*References.*–[68, 162].

**Family** *"Prohemerobiidae"* Handlirsch, 1906 [115]

gen. et sp. indet.

*Assigned material.*–BRSMG Cd 1397, isolated partial wing (9 mm long, 4.7 mm wide).

**Family** *incertae sedis*

gen. et sp. indet.

*Assigned material.*–Partial individual: NHMUK PI I.11409. Isolated wings: NHMUK PI I.66 & NHMUK PI I.66-83, part and counterpart; NHMUK PI I.11385. Isolated partial wings: NHMUK PI I.3337; NHMUK PI I.11326; NHMUK PI I.11401; NHMUK PI I.11463; NHMUK PI I.11464. Indeterminable: WARMS G 8087; NHMUK PI I.11434.

*Remarks.*–WARMS G 8087 was not available for re-examination but was reported as Neuroptera by Kelly [109]. See supplementary material for individual measurements and descriptions.

**Order** Diptera Linnaeus, 1758 [118]

**Suborder** Nematocera Latreille, 1825 [169]

**Family** Limoniidae Speiser, 1909 [170]

**Genus** *Architipula* Handlirsch, 1906 [115]

*Architipula anglicana* (Tillyard, 1933) [70]

(Fig 5G)

*Holotype.*–NHMUK PI I.11298, isolated wing (8.2 mm long, 2.2 mm wide).

*Assigned material.*–NHMUK PI I.11272, isolated wing (5.9 mm long, 2 mm wide); NHMUK PI I.11365, isolated wing (7.3 mm long, 2.5 mm wide); NHMUK PI I.11313, isolated partial wing (6.5 mm long, 2.6 mm wide); NHMUK PI I.11398, isolated partial wing (8.7 mm long, 1.3 mm wide).

*Remarks.*–Note that NHMUK PI I.11298 was formerly described within *Liassotipula* Tillyard, 1933 [70], which was then synonymized with *Architipula* by Kopeć *et al.* [76]. Venation of additional material is identical to other specimens attributed to *Architipula anglicana* as described by Kopeć *et al.* [76].

*References.*–[70, 76, 115].

*Architipula* sp.

*Assigned material.*–CHAGM F.633, isolated wing (6.2 mm long, 1.7 mm wide).

**Genus** *Grimmenia* Krzemiński and Zessin, 1990 [171]

*Grimmenia tillyardi* Kopeć *et al.*, 2017b [76]

(Fig 5H)

*Holotype.*–NHMUK PI I.3328, isolated wing (3.8 mm long, 0.9 mm wide).

*References.*–[76].

**Genus** *Mesotipula* Handlirsch, 1920 [160]

*Mesotipula slatteri* Kopeć *et al.*, 2018 [172]

*Holotype.*–NHMUK PI I.11316 & NHMUK PI I.11311, isolated female imago, part and counterpart (5.7 mm long, 1.9 mm wide).

*Assigned material.*–NHMUK PI I.3320 & NHMUK PI I.3323, isolated partial wing, part and counterpart (6.4 mm long, 1.7 mm wide).

*Remarks.*–Venation of additional material is identical to other specimens attributed to *Mesotipula slatteri* as described by Kopeć *et al.* [172].

*References.*–[172].

**Family** Limoniidae Speiser 1909 [170]

gen. et sp. indet

*Assigned material.*–Partial individual: NHMUK PI I.11329. Isolated skeletal elements: NHMUK PI I.71; NHMUK PI I.11328; NHMUK PI I.11348; NHMUK PI I.11380; NHMUK PI I.11391; NHMUK PI I.3319; NHMUK PI I.3345; NHMUK PI I.3350. Isolated partial wings: NHMUK PI In.59069; NHMUK PI In.59074; NHMUK PI In.59076.

**Family** *incertae sedis*

gen. et sp. indet

*Assigned material.*–NHMUK PI I.11276, isolated segmented body (5.3 mm long, 0.7 mm wide). Isolated partial wings: NHMUK PI I.73 (3.7 mm long, 1 mm wide); NHMUK PI I.11315 (7.9 mm long, 2.2 mm wide); NHMUK PI I.11358 (7.8 mm long, 0.5 mm wide).

Strawberry bank palaeoentomofauna

**Class** Insecta Linnaeus, 1758 [118]

**Subclass** Pterygota Brauer, 1885 [123]

**Order** Odonata Fabricius, 1793 [125]

**Suborder** Anisozygoptera

**Family** Liassogomphidae Tillyard, 1935 [130]

gen. et sp. indet.

*Assigned material.*–TTNCM 39/2011/0364, isolated wing fragment (10.7 mm long, 8 mm wide); TTNCM 39/2011/0509, isolated wing fragment (26.9 mm long, 7.9 mm wide).

*Remarks.*–TTNCM 39/2011/0509 is figured in Williams *et al.* [41] (Fig 3C).

*References.*–[41, 73].

**Family** *incertae sedis*

gen. et sp. indet.

*Assigned material.*–TTNCM 39/2011/0362a-b; TTNCM 39/2011/0403a-b; TTNCM 39/2011/0408; TTNCM 39/2011/0420a-b; TTNCM 39/2011/0503 & TTNCM 39/2011/0504 (part and counterpart); TTNCM 39/2011/0505; TTNCM 39/2011/0506; TTNCM 39/2011/0507a-b; TTNCM 39/2011/0511; TTNCM 39/2011/0512; TTNCM 39/2011/0514; TTNCM 39/2011/0515; TTNCM 39/2011/0516; TTNCM 39/2011/0517; TTNCM 39/2011/0528; TTNCM 39/2011/0545; TTNCM 39/2011/1013; TTNCM 39/2011/1014.

**Superorder** Dictyoptera Latreille, 1829 [133]

**Order** Blattodea Latreille, 1810 [114]

**Family** Caloblattinidae Vršanský and Ansorge, 2000 [173]

**Genus** *Rhipidoblattina* Handlirsch, 1906 [115]

*Rhipidoblattina* sp.

***Assigned material.***–TTNCM 39/2011/0537 & TTNCM 39/2011/0538, isolated forewing, part and counterpart (8.4 mm long, 2.8 mm wide).

***References.***–[173].

**Family** *incertae sedis*

gen. et sp. indet

***Assigned material.***–Isolated partial forewing: TTNCM 39/2011/1008. Isolated partial wings: TTNCM 39/2011/0391; TTNCM 39/2011/0392 & TTNCM 39/2011/0543 (part and counterpart); TTNCM 39/2011/0393 & TTNCM 39/2011/0395 (part and counterpart); TTNCM 39/2011/0396; TTNCM 39/2011/0398; TTNCM 39/2011/0405; TTNCM 39/2011/0410; plus an additional specimen with no accession number.

**Order** Dermaptera De Geer, 1773 [113]

**Family** Dermapteridae Vishnyakova, 1980 [174]

**Genus** *Trivenapteron* Kelly, Ross and Jarzembowski,2016 [42]

*Trivenapteron moorei* Kelly *et al.*, 2016 [42]

***Holotype.***–TTNCM 39/2011/0489, isolated sub-quadrate, truncate tegmen, pointed apically and without ornamentation.

***References.***–[41, 43].

**Order** Orthoptera Olivier, 1789 [117]

**Family** Locustopsidae Handlirsch, 1906 [115]

**Genus** *Locustopsis* Handlirsch, 1906 [115]

*Locustopsis* sp.

***Assigned material.***–NHMUK PI I.11485, isolated hindwing fragment (14 mm long, 2.4 mm wide).

***References.***–[115, 128, 138].

**Family** Locustopsidae Handlirsch, 1906 [115]

gen. et sp. indet.

***Assigned material.***–TTNCM 39/2011/0520 & TTNCM 39/2011/0522, isolated wing fragment, part and counterpart fragment (16 mm long, 3.8 mm wide); TTNCM 39/2011/0527, isolated forewing fragment (16.5 mm long, 4.2 mm wide); TTNCM 39/2011/0731, isolated wing (13 mm long, 2.6 mm wide); CAMSM TN 3200, partial wing (7.2 mm long, 2.8 mm wide).

**Family** Elcanidae Handlirsch, 1906 [115]

gen. et sp. indet.

***Assigned material.***–Isolated forewing: TTNCM 39/2011/0523. Isolated partial forewing: TTNCM 39/2011/0424a-b; TTNCM 39/2011/0525. Isolated wing: TTNCM 39/2011/0544. Isolated partial wing: TTNCM 39/2011/0732; TTNCM 39/2011/0411; TTNCM 39/2011/0397; TTNCM 39/2011/0419; TTNCM 39/2011/0434; TTNCM 39/2011/0415. Indeterminable: TTNCM 39/2011/0530; TTNCM 39/2011/0533; TTNCM 39/2011/0486a-b.

***Remarks.***–See supplementary material for individual measurements and descriptions. TTNCM 39/2011/0523 is figured in Williams *et al.* [41] (Fig 3D).

***References.***–[41, 115, 128, 142].

**Family** Protogryllidae Zeuner, 1937 [71]

gen. et sp. indet.

***Assigned material.***–TTNCM 39/2011/1006

***Remarks.***–Specimen was not available to be re-examined. TTNCM 39/2011/1006 was originally reported as belonging to Protogryllidae by Andrew J. Ross in the database he originally compiled on the specimens from TTNCM.

**Family** *incertae sedis*

gen. et sp. indet.

***Assigned material.*–**NHMUK PI I.11490; TTNCM 39/2011/0389; TTNCM 39/2011/0401; TTNCM 39/2011/0409; TTNCM 39/2011/0421; TTNCM 39/2011/0521; TTNCM 39/2011/ 0524; TTNCM 39/2011/0526; TTNCM 39/2011/0529; TTNCM 39/2011/0531; TTNCM 39/ 2011/0532a-b; TTNCM 39/2011/0534; TTNCM 39/2011/0552; TTNCM 39/2011/1007.

***Remarks.*–**See supplementary material for individual measurements and descriptions.

**Order** Hemiptera Linnaeus, 1758 [118]

**Suborder** Fulgoromorpha Evans, 1946 [146]

**Family** Fulgoridiidae Handlirsch, 1939 [128]

**Genus** *Margaroptilon* Handlirsch, 1906 [115]

*Margaroptilon brodiei* Handlirsch, 1906 [115]

***Holotype.*–**NHMUK PI I.3561, isolated forewing. Whitby Mudstone Formation: Dumbleton Member; Early Jurassic: Lower Toarcian of Alderton Hill, Gloucestershire, UK.

***Assigned material.*–**TTNCM 39/2011/0593, isolated forewing (5 mm long, 2.4 mm wide); NHMUK PI I.11492, isolated forewing (unable to be measured).

***Remarks.*–**See Alderton Hill remarks under *Margaroptilon* for discussion on *M. brodiei*.

***References.*–**[46, 109, 115].

*Margaroptilon* sp.

***Assigned material.*–**Isolated spotted wings: TTNCM 39/2011/0535 (8 mm long, 2.2 mm wide); TTNCM 39/2011/0592 (6.6 mm long, 2 mm wide); TTNCM 39/2011/0594 (5 mm long, 2 mm wide); TTNCM 39/2011/0595 (7.8 mm long, 2.2 mm wide); Isolated spotted wing fragment: TTNCM 39/2011/0995.

***Remarks.*–**Williams *et al.* [41] identified and figured TTNCM 39/2011/0594 as *Fulgoridulum* sp. (Fig 3A), however here we attribute this specimen to *Margaroptilon* sp.

**Suborder** Heteroptera Latreille, 1810 [114]

**Family** Archegocimicidae Handlirsch, 1906 [115]

***Assigned material.*–**TTNCM 39/2011/0707a-b, isolated partial forewing (3.2 mm long, 2.2 mm wide); TTNCM 39/2011/0394, isolated wing (3.1 mm long, 1 mm wide); TTNCM 39/ 2011/0549, isolated wing (3.6 mm long, 1.4 mm wide).

***Remarks.*–**TTNCM 39/2011/0707a-b is figured in Williams *et al.* [41] (Fig 3E).

**Family** *incertae sedis*

gen. et sp. indet.

***Assigned material.*–**Almost complete individuals: TTNCM 39/2011/0692; TTNCM 39/ 2011/0694; TTNCM 39/2011/0689a-b. Partial individual: TTNCM 39/2011/0665; TTNCM 39/ 2011/0693; TTNCM 39/2011/0696; NHMUK PI I.11493. Isolated bodies: TTNCM 39/2011/ 0646; TTNCM 39/2011/0648. Isolated skeletal elements: TTNCM 39/2011/0440; TTNCM 39/ 2011/0563a-b; TTNCM 39/2011/0570; TTNCM 39/2011/0647. Isolated partial forewings: TTNCM 39/2011/0700a-b; NHMUK PI I.11489. Isolated wing: TTNCM 39/2011/0416; TTNCM 39/2011/0429; TTNCM 39/2011/0480a-b; TTNCM 39/2011/0547; TTNCM 39/2011/ 0572; TTNCM 39/2011/0540a-b; TTNCM 39/2011/0562a-b. Isolated partial wing: TTNCM 39/2011/0399; TTNCM 39/2011/0431; TTNCM 39/2011/0541; TTNCM 39/2011/0542; NHMUK PI I.11488. Indeterminable: TTNCM 39/2011/0452; TTNCM 39/2011/0585; TTNCM 39/2011/0635; TTNCM 39/2011/1004.

**Order** Coleoptera Linnaeus, 1758 [118]

**Family** Cupedidae Laporte, 1836 [175]

gen. et sp. indet

***Assigned material.*–**TTNCM 39/2011/0624, isolated forewing (6.3 mm long, 1.7 mm wide); TTNCM 39/2011/0712, isolated partial forewing (11.5 mm long, 2.1 mm wide).

**Family** *incertae sedis*
gen. et sp. indet

***Assigned material.***–Almost complete individuals: TTNCM 39/2011/0365; TTNCM 39/2011/0377; TTNCM 39/2011/0640; TTNCM 39/2011/0643; TTNCM 39/2011/0644; TTNCM 39/2011/0645; TTNCM 39/2011/0666; TTNCM 39/2011/0667; TTNCM 39/2011/0697 & TTNCM 39/2011/0698 (part and counterpart); TTNCM 39/2011/0709; CAMSM TN 3199. Partial individuals: TTNCM 39/2011/0366; TTNCM 39/2011/0374; TTNCM 39/2011/0375; TTNCM 39/2011/0376; TTNCM 39/2011/0378; TTNCM 39/2011/0379; TTNCM 39/2011/0380; TTNCM 39/2011/0381; TTNCM 39/2011/0382; TTNCM 39/2011/0597; TTNCM 39/2011/0622; TTNCM 39/2011/0663 & TTNCM 39/2011/0664 (part and counterpart); TTNCM 39/2011/0669; TTNCM 39/2011/0670; TTNCM 39/2011/0671; TTNCM 39/2011/0672; TTNCM 39/2011/0673; TTNCM 39/2011/0708; TTNCM 39/2011/0710; TTNCM 39/2011/0711; NHMUK PI I.11486; NHMUK PI I.11487; NHMUK PI I.11497; NHMUK PI I.11500; CAMSM TN 3202. Isolated bodies: TTNCM 39/2011/0726; TTNCM 39/2011/0725. Isolated skeletal elements: TTNCM 39/2011/0441; TTNCM 39/2011/0443; TTNCM 39/2011/0471; TTNCM 39/2011/0496; TTNCM 39/2011/0662; TTNCM 39/2011/0695. Isolated forewings: TTNCM 39/2011/0363; TTNCM 39/2011/0367; TTNCM 39/2011/0386; TTNCM 39/2011/0387; TTNCM 39/2011/0388; TTNCM 39/2011/0406; TTNCM 39/2011/0471; TTNCM 39/2011/0596; TTNCM 39/2011/0598; TTNCM 39/2011/0599; TTNCM 39/2011/0600; TTNCM 39/2011/0601; TTNCM 39/2011/0602; TTNCM 39/2011/0603; TTNCM 39/2011/0604; TTNCM 39/2011/0605; TTNCM 39/2011/0606; TTNCM 39/2011/0607; TTNCM 39/2011/0608; TTNCM 39/2011/0609; TTNCM 39/2011/0610; TTNCM 39/2011/0613; TTNCM 39/2011/0614; TTNCM 39/2011/0615; TTNCM 39/2011/0616; TTNCM 39/2011/0617; TTNCM 39/2011/0620; TTNCM 39/2011/0621; TTNCM 39/2011/0627; TTNCM 39/2011/0630; TTNCM 39/2011/0631; TTNCM 39/2011/0668; TTNCM 39/2011/0713; TTNCM 39/2011/0714; TTNCM 39/2011/0715; TTNCM 39/2011/0717; TTNCM 39/2011/0718; TTNCM 39/2011/0719; TTNCM 39/2011/0720; TTNCM 39/2011/0721; TTNCM 39/2011/0722; TTNCM 39/2011/0723; TTNCM 39/2011/0724; TTNCM 39/2011/0727; TTNCM 39/2011/0728; TTNCM 39/2011/0760a-b; TTNCM 39/2011/0998; TTNCM 39/2011/1002; NHMUK PI I.11484; NHMUK PI I.11491; NHMUK PI; I.11495; NHMUK PI I.11496 & NHMUK PI I.11494 (part and counterpart); NHMUK PI I.11498; NHMUK PI I.11502; CAMSM TN 3203; CAMSM 3204. Isolated partial forewings: TTNCM 39/2011/0373; TTNCM 39/2011/0407; TTNCM 39/2011/0493; TTNCM 39/2011/0498; TTNCM 39/2011/0611; TTNCM 39/2011/0612; TTNCM 39/2011/0618; TTNCM 39/2011/0619; TTNCM 39/2011/0623; TTNCM 39/2011/0625; TTNCM 39/2011/0628 & TTNCM 39/2011/0629 (part and counterpart); TTNCM 39/2011/0716; TTNCM 39/2011/0949. Indeterminable: TTNCM 39/2011/0447; TTNCM 39/2011/0448; TTNCM 39/2011/0449; TTNCM 39/2011/0450; TTNCM 39/2011/0451; TTNCM 39/2011/0452; TTNCM 39/2011/0453; TTNCM 39/2011/0636; TTNCM 39/2011/0641a-b; TTNCM 39/2011/0657; TTNCM 39/2011/0658; TTNCM 39/2011/0659; TTNCM 39/2011/0660; TTNCM 39/2011/0661; TTNCM 39/2011/0674; TTNCM 39/2011/0675; TTNCM 39/2011/0676; TTNCM 39/2011/0677; TTNCM 39/2011/0678; TTNCM 39/2011/0679; TTNCM 39/2011/0680 & TTNCM 39/2011/0681 (part and counterpart); TTNCM 39/2011/0682; TTNCM 39/2011/0683; TTNCM 39/2011/0684a-b; TTNCM 39/2011/0685; TTNCM 39/2011/0686; TTNCM 39/2011/0687; NHMUK PI I.11501.

***Remarks.***–TTNCM 39/2011/0640 is figured in Williams *et al.* [41] (Fig 3B). At least two elytra morphotypes can be identified, including smooth (TTNCM 39/2011/0727) and striated elytra (TTNCM 39/2011/0614), indicating that the beetle fauna of Strawberry Bank was likely to consist of several species. Other beetle elements consist of well-preserved partial bodies (TTNCM 39/2011/0644).

**Superorder** Amphiesmenoptera Kiriakoff, 1948 [176]

**Order** *incertae sedis*

***Assigned material.***–TTNCM 39/2011/0996a-b, isolated partial wing (9 mm long, 2.2 mm wide); TTNCM 39/2011/1009, isolated partial wing (6.5 mm long, 3 mm wide); TTNCM 39/2011/1010, isolated partial wing (5.2 mm long, 3 mm wide).

**Order** Necrotrichoptera Engel, 2022 [152]

**Family** Necrotauliidae Handlirsch, 1906 [115]

**Genus** *Necrotaulius* Handlirsch, 1906 [115]

*Necrotaulius parvulus* Geinitz, 1884 [66]

***Holotype.***–FGWG 119/7, isolated wing. Grimmen Formation, 'Green Series'; Early Jurassic: Lower Toarcian of Dobbertin, Mecklenburg-Vorpommern, Germany.

***Assigned material.***–TTNCM 39/2011/0733, isolated wing (7.3 mm long, 3 mm wide).

***Remarks.***–See above for the remarks and discussion of *N. parvulus* for the Alderton Hill specimens.

***References.***–[43, 66, 77].

**Order** Mecoptera Packard, 1886 [157]

**Family** Orthophlebiidae Handlirsch, 1906 [115]

***Assigned material.***–TTNCM 39/2011/0400, isolated wing (9.9 mm long, 2.2 mm wide).

**Family** *incertae sedis*

gen. et sp. indet.

***Assigned material.***–TTNCM 39/2011/0546, isolated wing (6.1 mm long, 3.4 mm wide).

**Order** Diptera Linnaeus, 1758 [118]

**Family** Limoniidae Speiser, 1909 [170]

**Genus** *Architipula* Handlirsch, 1906 [115]

*Architipula* sp.

***Assigned material.***–TTNCM 39/2011/0402, isolated wing (7.1 mm long, 1.9 mm wide).

**References.**–[41, 76].

**Order** Neuroptera Linnaeus, 1758 [118]

**Family** *incertae sedis*

gen. et sp. indet.

***Assigned material.***–TTNCM 39/2011/0423; TTNCM 39/2011/0432; TTNCM 39/2011/0433; TTNCM 39/2011/0553; TTNCM 39/2011/1011; TTNCM 39/2011/1012.

**Superorder** Panorpida Hinton, 1958 [177]

**Family** *incertae sedis*

gen. et sp. indet.

***Assigned material.***–Isolated wing: TTNCM 39/2011/0539. Isolated partial wings: TTNCM 39/2011/0368; TTNCM 39/2011/0372; TTNCM 39/2011/0390; TTNCM 39/2011/0404; TTNCM 39/2011/0412; TTNCM 39/2011/0418; TTNCM 39/2011/0425; TTNCM 39/2011/0428; TTNCM 39/2011/0430; TTNCM 39/2011/0436; TTNCM 39/2011/0437; TTNCM 39/2011/0548.

***Remarks.***–See supplementary material for individual measurements and descriptions.

## Palaeoentomofaunal overview

At order level, the individual-based rarefaction curves for Alderton Hill and Strawberry Bank show that counts from both localities are fully saturated, indicating that they are both well-sampled (Fig 6A). At family level, the rarefaction curve for Alderton Hill is approaching saturation, showing that this assemblage is reasonably well-sampled; whereas the curve at Strawberry Bank continues to increase, showing that this assemblage is incompletely sampled at this

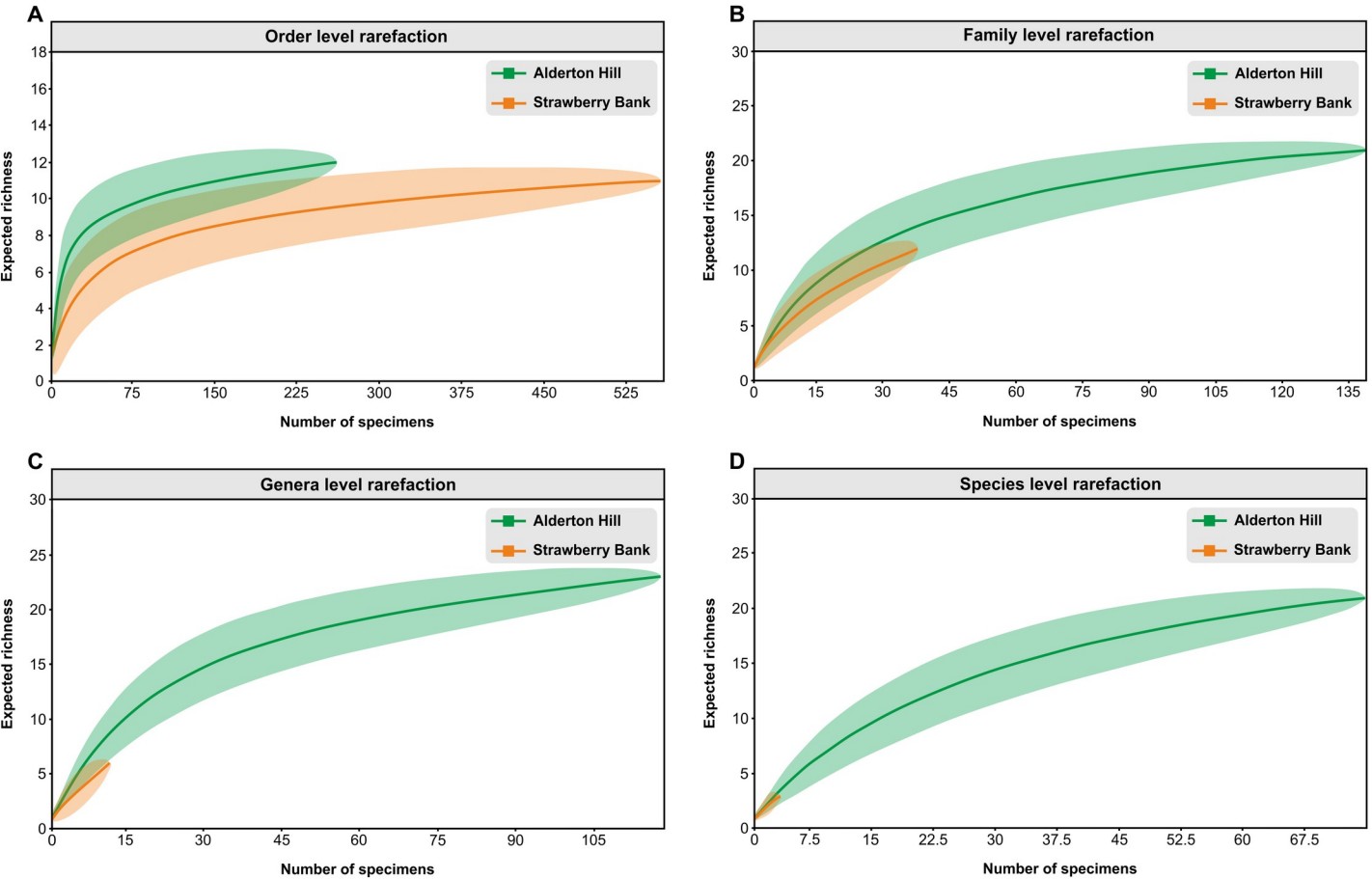

**Fig 6.** Individual-based rarefaction curves for Alderton Hill and Strawberry Bank at the following taxonomic ranks: (**A**) order level; (**B**) family level; (**C**) genus level; (**D**) species level. The rarefaction curves show the expected richness (y-axis) for a given number of specimens (x-axis). The green and orange curves represent the Alderton Hill and Strawberry Bank insect assemblages respectively. Shaded areas represent 95% confidence intervals.

taxonomic level (Fig 6B). At genus level, similar to family level, the rarefaction curves show that Alderton Hill is reasonably well-sampled, and that Strawberry Bank is incompletely sampled (Fig 6C). At species level, the rarefaction curves for both Alderton Hill and Strawberry Bank do not approach saturation (Fig 6D), showing that the insect assemblages are incompletely sampled, and are therefore not a good indication of species richness. Additionally, qualitative inspection of all rarefaction curves at each taxonomic level indicate that taxonomic richness at Alderton Hill is higher than that at Strawberry Bank (Fig 6A). Overall, this analysis shows that the Strawberry Bank assemblage is incompletely sampled in terms of diversity below the order level, whereas at Alderton Hill the palaeoentomofauna can be considered well-sampled at order level and reasonably well-sampled at family and genus levels.

## Alderton Hill

From the 370 individual insects documented and examined from Alderton Hill, 29.2% (n = 108; Table 4) of the total assemblage was considered indeterminate due to poor preservation and therefore could not be confidently assigned to any taxonomic group; these specimens were not included within further taxonomic analysis. The remaining 70.8% (n = 262) could be taxonomically identified to at least the level of order (Table 4). Of these individuals, 37.6%

**Table 4. Number and percentage of Toarcian insect specimens attributed to each taxonomic rank at both localities at Alderton Hill and Strawberry Bank.** Percentages are given as a proportion of the total number of individuals.

| Assemblage breakdown | Alderton Hill insect assemblage No. (%) | Strawberry Bank insect assemblage No. (%) |
|---|---|---|
| **Number of specimens** (i.e. assigned an accession number) | 363 | 386 |
| **Number of individuals** (i.e. multiple individuals on one specimen) | 370 | 799 |
| **Number of indeterminate individuals** | 108 (29.2) | 240 (30) |
| Number of individuals assigned to order[a] | 262 (70.8) | 559 (70) |
| Number of insect **orders**[a] identified | 12 | 11 |
| Number of individuals assigned to **family** | 139 (37.6) | 38 (4.8) |
| Number of insect **families** identified | 21 | 12 |
| Number of individuals assigned to **genera** | 118 (31.9) | 12 (1.5) |
| Number of insect **genera** identified | 23 | 6 |
| Number of individuals assigned to **species** | 75 (20.3) | 4 (0.5) |
| Number of insect **species** identified | 21 | 3 |

[a]The total number of insect orders for Strawberry Bank also includes the superorders Amphiesmenoptera and Panorpida.

(n = 139) could be further assigned to family level, whilst 31.9% (n = 118) and 20.3% (n = 75) could be further assigned to genera and species respectively (Table 4). The 12 insect orders identified from the 262 identifiable individuals collected at Alderton Hill are Hemiptera (true bugs), Coleoptera (beetles), Orthoptera (grasshoppers), Neuroptera (lacewings), Odonata (dragonflies), Diptera (flies), Mecoptera (scorpionflies), Blattodea (cockroaches), Necrotrichoptera (caddisflies), Reculida ('winged insects'),? Dermaptera (earwigs) and Ephemeroptera (mayflies); and their relative abundance is shown in Fig 7A.

The Hemiptera of Alderton Hill are the most diverse and abundant order within the fossil insect assemblage, comprising 22.1% (n = 58; Fig 7A) of all the identifiable insect specimens collected from this locality. This order is represented by four families (Table 5). The majority of the hemipteran material, in particular those attributed to Fulgoridiidae, are comprised of isolated wings. The genus *Margaroptilon* is the most common taxon within this order, and is represented by well-preserved forewings with spotted pigmentation (i.e. NHMUK PI I.11266; Fig 4C). Other hemipteran elements also include partial bodies (Figs 3I–4G), which cannot be identified further due to their poor preservation. Coleoptera is the second most common insect order of the Alderton Hill fossil assemblage (20.6%; n = 54). This order is mostly represented by isolated elytra, but also paired elytra and partial individuals, and the poor preservation prevents further identification (Table 5). However, at least three elytra morphotypes can be identified (Fig 4K–Fig 4M), indicating that the beetle fauna is likely to consist of several genera or species. Orthoptera (13.4%, n = 35; Fig 7A) are attributed to three families (Table 5) and are primarily represented by fragmented, yet well-preserved tegmina. Some tegmina possess the original striped and spotted pigmentation (Fig 3I–3L), which aids their taxonomic assignment. Isolated skeletal elements including saltatorial hind legs are also present (Fig 3M). Whilst Neuroptera are relatively abundant from Alderton Hill (12.2%, n = 32; Fig 7A), they are

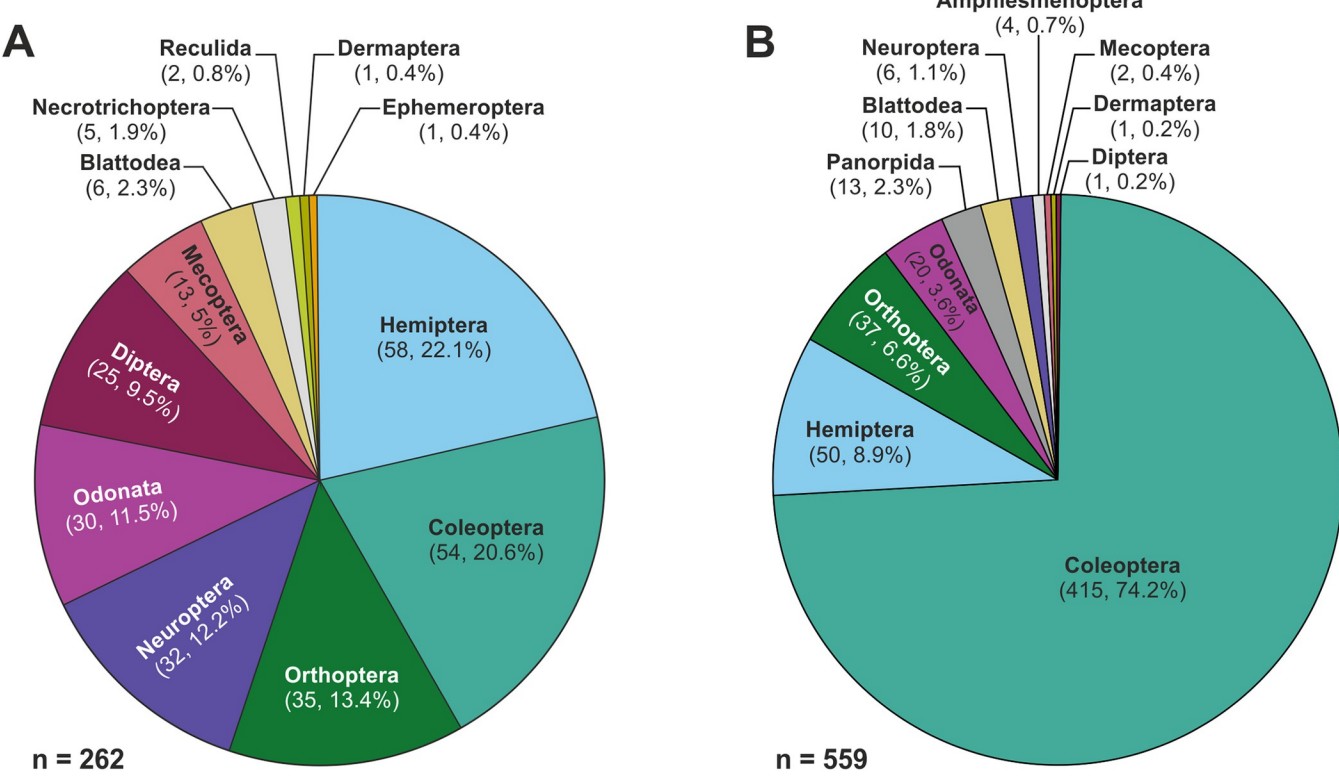

**Fig 7.** Pie-charts showing the number and percentages of taxonomically identifiable insect specimens within each order, grouped by abundance from (**A**) Alderton Hill, Gloucestershire and (**B**) Strawberry Bank, Somerset: Note that the 'winged insects' Amphiesmenoptera and Panorpida are superorders, but are still included within this comparison.

mostly represented by poorly preserved isolated wings. However, several well-preserved examples of Neuroptera are present (Fig 5A–5F), representing three families (Table 5).

Odonata is represented by three families (Table 5) and accounts for 11.5% (n = 30) of the total Alderton Hill palaeoentomofauna (Fig 7A). The order is mainly represented by partial wing fragments that possess clear venation (Fig 3C and 3D) aiding identification, however complete, well-preserved forewings (Fig 3A) and hindwings (Fig 3B) are also present. Diptera comprises 9.5% (n = 25) of the fossil assemblage (Fig 7A), in which all specimens identified to family level are exclusively assigned to the family Limoniidae (craneflies) and primarily composed of isolated bodies and wings. Well-preserved isolated wings belonging to the order Mecoptera comprises 5% (n = 13) of the total insect assemblage (Figs 3R and 4Q; Fig 6A), and there are three families present. Blattodea comprise 2.3% (n = 6) (Fig 7A) and are represented by rather well-preserved, isolated tegmina with the clavus attached (Fig 3G–3H). One specimen, BRSMG Cg2374 & BRSMG Cg2375, is noticeably dissimilar to all other Toarcian cockroaches and therefore represents a new genus and species. Necrotrichoptera accounts for only 1.9% (n = 5) of the Alderton Hill palaeoentomofauna (Fig 7A), in which all specimens belong to the family Necrotauliidae (Fig 4N and 4O). Similarly, Reculida is also uncommon (0.8%; n = 2) (Fig 7A) and is only represented by partial, yet well preserved forewing fragments of *Geinitzia carpentieri* (Fig 3F).? Dermaptera (NHMUK PI I.11403; Fig 3E) and Ephemeroptera (WARMS G 8084) are rare orders within the deposit, each representing 0.4% (n = 1) of the total insect assemblage (Fig 7A); these orders are all represented by poorly preserved fragmentary remains.

**Table 5. Number and percentage of specimens attributed to each family, genera and species within each order at Alderton Hill.** Taxa are listed in the order they appear within the catalogue of identified taxa. Percentages are given as a proportion of the total number of individuals attributed to each order.

| Order | Family | Genera and species | No. of assigned specimens | Percentage of taxa within order (%) |
|---|---|---|---|---|
| Ephemeroptera | - | - | 1 | 100 |
| Odonata | Heterophlebiidae | *Heterophlebia buckmani* | 7 | 23.3 |
| | Liassogomphidae | *Heterothemis brodiei* | 6 | 20 |
| | | *Heterothemis* sp. | 4 | 13.3 |
| | Protomyrmeleontidae | *Protomyrmeleon* sp. | 1 | 3.3 |
| | | gen. et sp. Indet. | 1 | 3.3 |
| | *incertae sedis* | gen. et sp. Indet. | 11 | 36.8 |
| ?Dermaptera | - | - | 1 | 100 |
| Reculida | Geinitziidae | *Geinitzia carpentieri* | 2 | 100 |
| Blattodea | ?Raphidiomimidae | *Liadoblattina blakei* | 4 | 66.6 |
| | uncertain | undescribed | 1 | 16.7 |
| | *incertae sedis* | gen. et sp. Indet | 1 | 16.7 |
| Orthoptera | Locustopsidae | *Locustopsis* sp. | 7 | 20 |
| | | gen. et sp. Indet | 1 | 2.9 |
| | Elcanidae | "*Elcana*" cf. *geinitzi* | 4 | 11.4 |
| | | "*Elcana*" sp. | 2 | 5.7 |
| | Protogryllidae | *Protogryllus* cf. *dobbertinensis* | 1 | 2.9 |
| | | *Protogryllus magnus* | 1 | 2.9 |
| | | *Protogryllus* sp. | 2 | 5.7 |
| | *incertae sedis* | gen. et sp. Indet | 17 | 48.5 |
| Hemiptera | Fulgoridiidae | *Fulgoridium* sp. | 9 | 15.5 |
| | | *Margaroptilon brodiei* | 18 | 31 |
| | | gen. Et sp. Indet. | 2 | 3.4 |
| | Archijassidae | *Archijassus heeri* | 1 | 1.7 |
| | Progonocimicidae | - | 2 | 3.4 |
| | Gerromorpha | *Engynabis tenuis* | 1 | 1.7 |
| | *incertae sedis* | gen. Et sp. Indet. | 25 | 43.3 |
| Coleoptera | *incertae sedis* | gen. et sp. Indet. | 54 | 100 |
| Necrotrichoptera | Necrotauliidae | *Necrotaulius parvulus* | 4 | 80 |
| | | ?*Necrotaulius* sp. | 1 | 20 |
| Mecoptera | Bittacidae | *Protobittacus handlirschi* | 1 | 7.7 |
| | | gen. et sp. Indet. | 1 | 7.7 |
| | Orthophlebiidae | *Orthophlebia brodiei* | 1 | 7.7 |
| | | *Orthophlebia* sp. | 9 | 69.2 |
| | Pseudopoly-centropodidae | *Pseudopoly-centropus* sp. | 1 | 7.7 |
| Neuroptera | Osmylopsychopidae | *Actinophlebia intermixta* | 3 | 9.4 |
| | | *Actinophlebia* sp. | 1 | 3.1 |
| | "Familia nova A" | *Paractinophlebia curtisii* | 2 | 6.3 |
| | "Prohemerobiidae" | "*Prohemerobius*"*aldertonensis* | 3 | 9.4 |
| | | "*Prohemerobius*" sp. | 3 | 9.4 |
| | | "*Archeosmylus*" *alysius* | 6 | 18.8 |
| | | "*Archeosmylus*" *complexus* | 2 | 6.3 |
| | | gen. et sp. Indet. | 1 | 3.1 |
| | *incertae sedis* | gen. et sp. Indet. | 11 | 34.2 |

*(Continued)*

**Table 5.** (Continued)

| Order | Family | Genera and species | No. of assigned specimens | Percentage of taxa within order (%) |
|---|---|---|---|---|
| **Diptera** | Limoniidae | *Architipula anglicana* | 5 | 20 |
| | | *Architipula* sp. | 1 | 4 |
| | | *Grimmenia tillyardi* | 1 | 4 |
| | | *Mesotipula slatteri* | 2 | 8 |
| | | gen. et sp. Indet. | 16 | 64 |

## Strawberry bank

Of the 799 individual insects from Strawberry Bank, 30% (n = 240; Table 4) of the insect assemblage are considered indeterminate due to poor preservation and therefore cannot be confidently assigned to any taxonomic group. Of the total insect assemblage, 70% (n = 559) can be taxonomically identified to at least order level (Table 4). Of these individuals, only 4.8% (n = 38) of the Strawberry Bank insect assemblage can be assigned to family level, whilst 1.5% (n = 12) and 0.5% (n = 4) can be assigned to genera and species (Table 4). Nine insect orders and two superorders are identified among 559 identifiable individuals from Strawberry Bank: Hemiptera, Coleoptera, Orthoptera, Neuroptera, Odonata, Diptera, Mecoptera, Blattodea, Dermaptera, Amphiesmenoptera and Panorpida; and their relative abundance is shown in Fig 7B.

Within the Strawberry Bank fossil insect assemblage, Coleoptera is the most abundant order comprising 74.2% (n = 415; Fig 7B) of all the identifiable insect specimens collected from this locality. The majority of the coleopteran material is represented by disarticulated and fragmented elytra that are compressed and rather badly preserved, and not possible to identify beyond order level (Table 6). Similar to Alderton Hill, at least two elytra morphotypes can be identified, indicating different species. Hemiptera is the second most common insect order of the Strawberry Bank palaeoentomofauna (8.9%, n = 50) (Fig 7B), and can be assigned to two families (Table 6). This order is primarily represented by partial individuals (NHMUK PI I.11493) and isolated wings; wings attributed to *Margaroptilon* (Table 6) in particular possess spotted pigmentation (TTNCM 39/2011/0594). Orthoptera, the third most abundant insect order (6.6%, n = 37) (Fig 7B), is represented by three families (Table 6) and comprised of isolated, fragmented tegmina, some of which show original striped and spotted pigmentation (TTNCM 39/2011/0523). Odonata accounts for 3.6% (n = 20) of the Strawberry Bank palaeoentomofauna (Fig 7B) and can be attributed to only one family (Table 6). This order is mainly represented by partial wing fragments that possess some venation (e.g. TTNCM 39/2011/0516). Although disarticulated, segmented abdominal elements (TTNCM 39/2011/0362a-b) are also documented. Panorpida (2.3%, n = 13; Fig 7B) is represented exclusively by isolated and often poorly preserved wing fragments (e.g. TTNCM 39/2011/0436), preventing taxonomic refinement below superorder level (Table 6).

Blattodea accounts for 1.8% (n = 10) of the palaeoentomofauna (Fig 7B), and is represented by only one family (Table 6). Similar to the rest of the Strawberry Bank insect assemblage, the material is preserved as disarticulated tegmina, with no hindwings present. Neuroptera comprises 1.1% (n = 6) of the Strawberry Bank fossil insect assemblage (Fig 7B) and is mostly represented by poorly preserved, isolated wings that are difficult to identify beyond order level (Table 6); however, some better preserved examples are present (TTNCM 39/2011/0423). Amphiesmenoptera comprises 0.7% (n = 4) of the palaeoentomofauna (Fig 7B), with one family identified (Table 6). Whilst the rest of these individuals cannot be taxonomically refined any further than superorder level, they most likely belong to either Lepidoptera or

**Table 6. Number and percentage of specimens attributed to each family, genera and species within each order at Strawberry Bank.** Taxa are listed in the order they appear within the catalogue of identified taxa. Percentages are given as a proportion of the total number of individuals attributed to each order.

| Order | Family | Genera and species | No. of assigned specimens | Percentage of taxa within order (%) |
|---|---|---|---|---|
| **Odonata** | Liassogomphidae | - | 2 | 10 |
| | *incertae sedis* | gen. et sp. Indet. | 18 | 90 |
| **Blattodea** | Caloblattinidae | *Rhipidoblattina* sp. | 1 | 10 |
| | *incertae sedis* | gen. et sp. Indet. | 9 | 90 |
| **Dermaptera** | Dermapteridae | *Trivenapteron moorei* | 1 | 100 |
| **Orthoptera** | Locustopsidae | *Locustopsis* sp. | 1 | 2.7 |
| | Elcanidae | - | 13 | 35.1 |
| | Protogryllidae | - | 1 | 2.7 |
| | *incertae sedis* | gen. et sp. Indet. | 22 | 59.5 |
| **Hemiptera** | Fulgoridiidae | *Margaroptilon brodiei* | 2 | 4 |
| | | *Margaroptilon* sp. | 5 | 10 |
| | Archegocimicidae | - | 3 | 6 |
| | *incertae sedis* | gen. et sp. Indet. | 40 | 80 |
| **Coleoptera** | Cupedidae | - | 2 | 0.5 |
| | *incertae sedis* | gen. et sp. Indet. | 413 | 99.5 |
| **Amphiesmenoptera** | Necrotauliidae | *Necrotaulius parvulus* | 1 | 25 |
| | *incertae sedis* | - | 3 | 75 |
| **Mecoptera** | Orthophlebiidae | - | 1 | 50 |
| | *incertae sedis* | gen. et sp. Indet. | 1 | 50 |
| **Diptera** | Limoniidae | *Architipula* sp. | 1 | 100 |
| **Neuroptera** | *incertae sedis* | gen. et sp. Indet. | 6 | 100 |
| **Panorpida** | - | - | 13 | 100 |

Necrotrichoptera. This material is represented exclusively by isolated, preserved wing and wing fragments (TTNCM 39/2011/1009). Whilst Dermaptera is uncommon within the Strawberry Bank palaeoentomofauna (0.2%, n = 1) (Fig 7B), a single tegmen was described by Kelly *et al*. [42] as a new species—*Trivenapteron moorei* (TTNCM 39/2011/0489), which was the first record of an earwig from the Toarcian of the UK [41]. Finally, Mecoptera and Diptera are rare within the Strawberry Bank deposit, and represent 0.4% (n = 2) and 0.2% (n = 1) of the total insect assemblage respectively (Fig 7B).

In summary, approximately 70% of the insect assemblages at both Alderton Hill and Strawberry Bank can be identified to order level, and whilst the Coleoptera, Hemiptera and Orthoptera are the three most common orders at both localities, their relative abundances differ (Fig 7). Alderton Hill is significantly more diverse at all taxonomic ranks recognised in the fossil assemblage (Table 3), as more specimens can be assigned to family (AH = 37.6% cf. SB = 4.8%), genera (AH = 31. 9% cf. SB = 1.5%) and species (AH = 20.3% cf. SB = 0.5% (Table 3).

## Feeding traits

The diet of 74.3% (n = 278) of the Alderton Hill fossil insect assemblage is classified as indeterminable. Specific diets for the majority of taxa cannot be inferred because of the low taxonomic resolution (e.g. Coleoptera). At present-day, carnivorous, herbivorous and parasitic examples are present in almost all insect orders; demonstrating that generalisations cannot be made. For taxa where their specific diets can be determined (n = 92), 58.7% (n = 54) are herbivorous (e.g. Hemiptera: Fulgoromorpha and Coleorrhyncha; Orthoptera: Elcanidae, Locustopsidae and

Protogryllidae) and 41.3% (n = 38) are carnivorous (e.g. Blattodea: Raphidiomimidae; Mecoptera: Bittacidae; and Odonata). However, the low numbers in each feeding trait mean that analysis is not very representative. As the majority of the Strawberry Bank palaeoentomofauna is comprised of Coleoptera, which are largely unidentifiable beyond order level, further insights in the trophic feeding of this insect community and specific diets cannot be inferred.

## Taphonomic analysis

### Alderton hill

Based on the matrix of the insect specimens examined, all of the insect material collected from Alderton Hill to date appears to have come from the limestone nodules within the Fish Bed (Bed 10 of Richardson [79]). The insect material of Alderton Hill is mostly preserved as disarticulated, fragmented compression fossils. Some Coleoptera are preserved in three-dimensions, but show minor compression (NHMUK PI I.11318; NHMUK PI I.11369). As the insects are preserved in early diagenetic nodules formed within fine-grained sedimentary deposits, the intricate detail of the insects is well preserved, including wing venation and evidence of colour pigmentation and patterns.

From the 370 individual insects documented and examined, 20.5% (n = 76) could not be taphonomically assigned because of poor preservation or, in a few cases, they were not available for examination. These specimens were assigned the taphonomic grade '??'. Of the taxonomically identifiable specimens, 56.1% (n = 147) of the total insect assemblage were comprised of heavily sclerotised taxa i.e. Hemiptera, Coleoptera and Orthoptera. No aquatic larval stages were observed within the collections, and thus all of the specimens examined were represented by terrestrial adult life stages, including flying terrestrial and semiaquatic taxa.

The taphonomic grades from 1 – 2d (Table 2) were assigned to 294 individuals (Fig 8A). Whilst a wide range of preservation was observed among the Alderton Hill insects (Table 3), the material was mostly represented by isolated wings (n = 198, 67.3%; Fig 8A). The most common primary taphonomic grade within the deposit was isolated forewing/s (grade 2b) with 101 (34.4%) individuals; 50 (17%) and 51 (17.3%) of these forewings could be assigned to the secondary taphonomic grades 2b.1 and 2b.2 respectively (Fig 8A). Eighty-six individuals (29.3%) were isolated unassigned wing/s (grade 2d), with 12 (4.1%) being almost complete (grade 2d.1) and 74 (25.2%) wing fragments (grade 2d.2) (Fig 8A); the latter was the most common secondary taphonomic grade recorded from the insect assemblage. Seventy-one individuals (24.1%) were isolated bodies (grade 2a); 12 (4.1%) and 59 (20.1%) were almost complete (grade 2a.1) and skeletal body elements (grade 2a.2) (Fig 8A). Twenty-five individuals (8.5%) were bodies with forewings and/or hindwings (grade 1); 7 (2.4%) insects were assigned to the secondary taphonomic grade 1.2, and 18 (6.1%) individuals to 1.3 (Fig 8A). No complete, fully articulated individuals (grade 1.1) have been collected from this locality to date (Fig 8A). One of the most complete insect collected from Alderton Hill was the paratype of *Heterophlebia buckmani* (NHMUK PI I.3988), comprised of a head, thorax and fore- and hindwings, with only the distal half of abdomen absent [63]; however, this type specimen was not located within the NHMUK collection during this study and is presumed lost. Whilst this specimen could not be included within this research as it could not be re-examined, it demonstrates that more complete individuals were preserved at Alderton Hill.

The rarest primary taphonomic grade within the deposit was isolated hindwing/s (grade 2c) with 11 individuals (3.7%); 4 (1.4%) of these hindwings were almost complete (grade 2c.1) and 7 (2.4%) were fragments (grade 2c.2) (Fig 8A). The level of disarticulation is high in all insect orders; with only 8.5% (n = 25) of the total assemblage being considered either partially

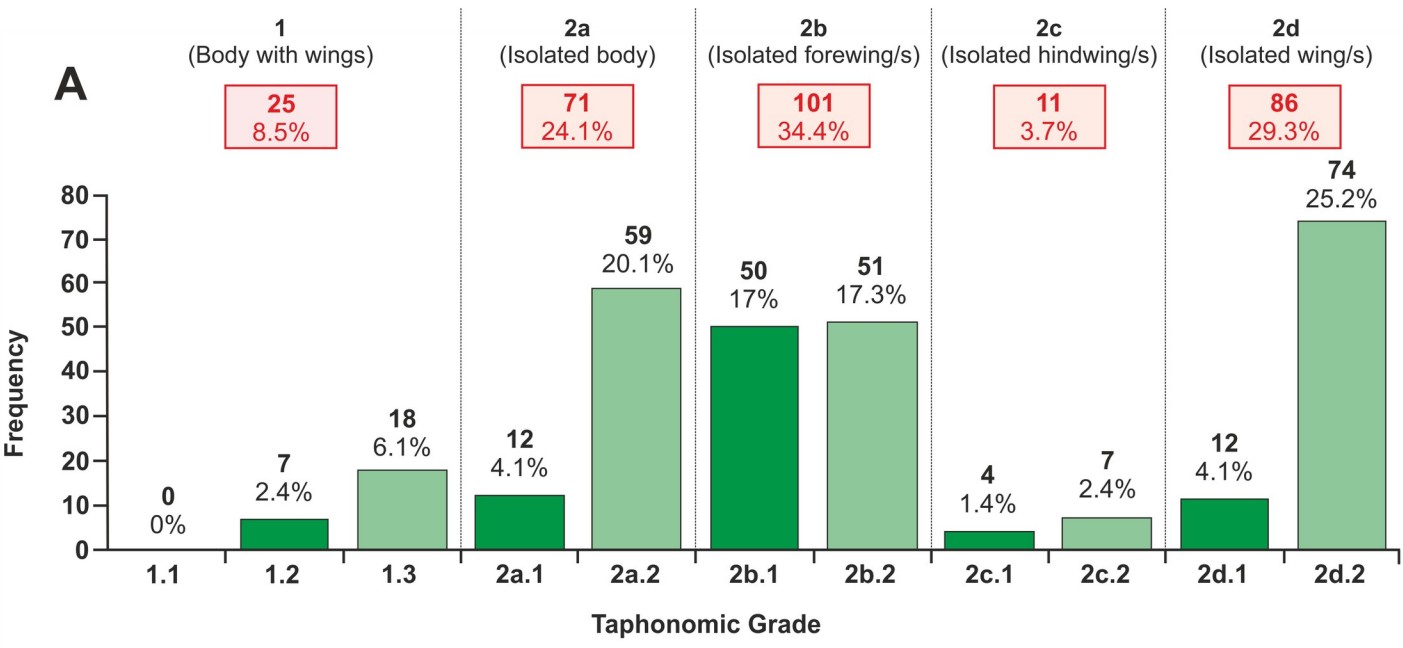

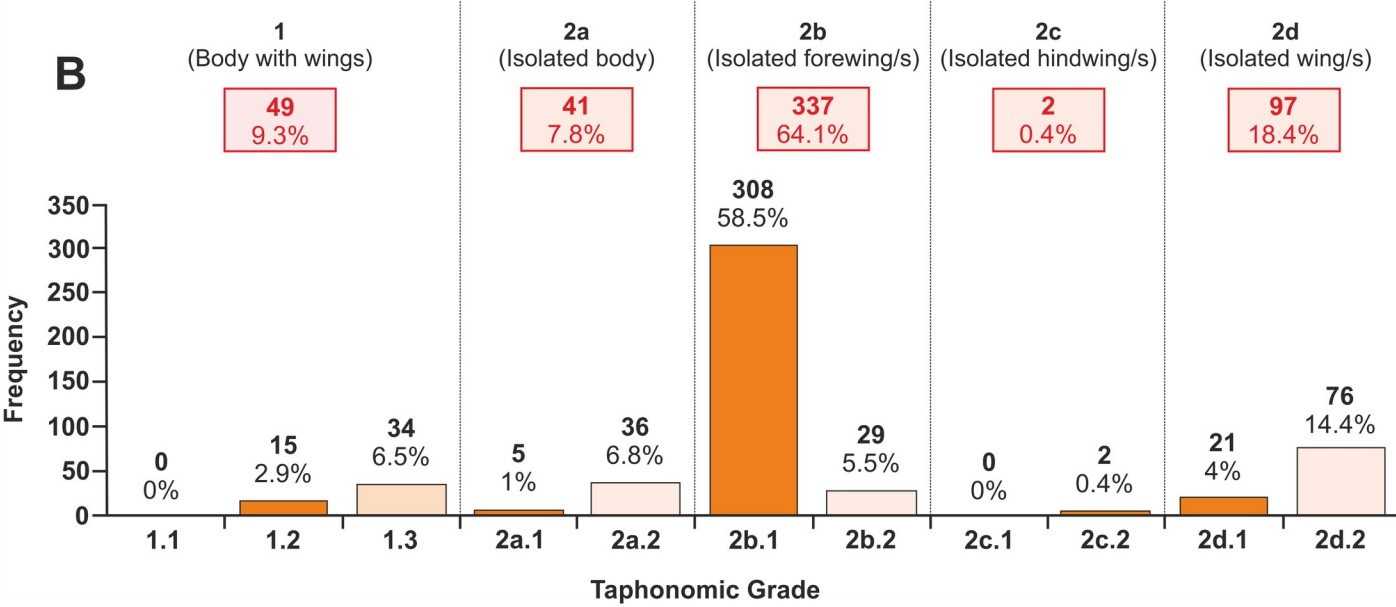

**Fig 8.** The frequency of individual insect specimens in each taphonomic grade for: **(A)** Alderton Hill, Gloucestershire (n = 291); **(B)** Strawberry Bank, Somerset (n = 518). The number and percentage in each primary taphonomic grades are given in red boxes, and the secondary taphonomic grade counts and percentages are stated above each individual bar. The solid green and orange bars denote complete specimens or element, and the paler green and orange bars denote partial specimens or elements.

articulated or fully articulated (e.g. abdomen associated with elytra and thorax); therefore, much of the Alderton Hill collection consisted of fully disarticulated insect elements (n = 269).

In total, 230 (78.2%) of the individuals assigned a taphonomic grade could be taxonomically assigned to at least order level (Table 3). A Chi-squared test indicated that there was a statistically significant association between the taxonomic rank and the taphonomic grade at order

level ($x^2$ = 234.58, df = 99, p = < 0.001). The excellent preservation of bodies with wings (grade 1), isolated complete bodies (grade 2a.1) and wings >75% complete (i.e. grades 2b.1 and 2c.1) in the majority of the specimens allowed for a high level of taxonomic identification. Where the wings were less than 75% complete (i.e. grades 2b.2, 2c.2 and 2d.2) or there were only isolated skeletal elements (grade 2a.2), the taxonomic identification was poorer. In order to determine the degree to which each primary taphonomic grade contributed to the total Chi-squared score at order level, Pearson residuals were calculated for each entry in the Chi-squared contingency table and displayed graphically (Fig 9). Within the individual orders, the largest range of taphonomic grades were observed within the Hemiptera, with 30% (n = 15) of this material being assigned to 2b.2. Coleoptera were primarily represented by isolated or paired elytra, as 45.5% (n = 20) of the coleopteran material was assigned to 2b.1 (Fig 9).a

The Alderton Hill insect fossils had a minimum preserved length of 0.3 mm and a maximum of 45 mm, with a median preserved length of 6.4 mm and a mean preserved length of 7.98 mm (SD = 6.10 mm; n = 318) (Fig 10A). Preserved specimen widths ranged from 0.1 mm to 13.9 mm, with a median preserved width of 2.3 mm and a mean preserved width of 2.84 mm (SD = 2.12 mm; n = 318) (Fig 10C); only 5% of the insect material was greater than 20 mm long and 11.9% was greater than 5 mm wide.

## Strawberry bank

Similar to Alderton Hill, the levels of disarticulation were high in all insect orders at Strawberry Bank; only 9.3% (n = 49) of the total insect assemblage was considered either partially articulated or fully articulated, therefore much of the deposit consisted of fully disarticulated insect elements (n = 477) (Fig 8B).

As a high proportion (74.2%) of the specimens from Strawberry Bank are taphonomically robust Coleoptera, most of the material is disarticulated but preserved in three-dimensions, with minor amounts of compression. The remainder of the material is preserved as highly disarticulated, and fragmented compression fossils, similar to Alderton Hill. From the 799 individual insects documented and examined from the Strawberry Bank insect assemblage, 34.2% (n = 273) could not be taphonomically assigned and were given the taphonomic grade '??'. Of the taxonomically identifiable specimens, 89.8% (n = 502) of the total insect assemblage of Strawberry Bank was comprised of heavily sclerotised taxa i.e. Coleoptera, Hemiptera and Orthoptera. The taphonomic grades from 1–2d (Table 2) were assigned to 526 individuals (Fig 8B).

Similar to Alderton Hill, the insect material of Strawberry Bank was mostly represented by isolated wings (n = 436, 82.9%; Fig 8B). The commonest primary taphonomic grade within the deposit was isolated forewing/s (grade 2b) with 337 (64.1%) individuals. From this, 308 (58.6%) of these forewings were almost complete (grade 2b.1) and 29 (5.5%) were fragments (grade 2b.2) (Fig 8B). The former was the most common secondary taphonomic grade recorded from the Strawberry Bank insect assemblage.

Ninety-seven individuals (18.4%) were isolated unassigned wing/s (grade 2d), with 21 (4%) and 76 (14.4%) of these wings belonging to the secondary taphonomic grades 2d.1 and 2d.2 respectively (Fig 8B). Forty-nine individuals (9.3%) were bodies with forewings and/or hindwings (grade 1); 15 (2.9%) insects were assigned to the secondary taphonomic grade 1.2, and 34 (6.5%) individuals to 1.3 (Fig 8B). Forty-one individuals (7.8%) were isolated bodies (grade 2a); 5 (1%) and 36 (6.8%) were given the secondary taphonomic grades 2a.1 and 2a.2 respectively (Fig 8B). No insects could be assigned to taphonomic grade 1.1 (Fig 8B), meaning no complete, fully articulated individuals have been recovered from Strawberry Bank to date. The rarest primary taphonomic grade within this deposit was isolated hindwing/s (grade 2c) with only 2 individuals (0.4%) assigned to taphonomic grade 2c.1 (Fig 8B).

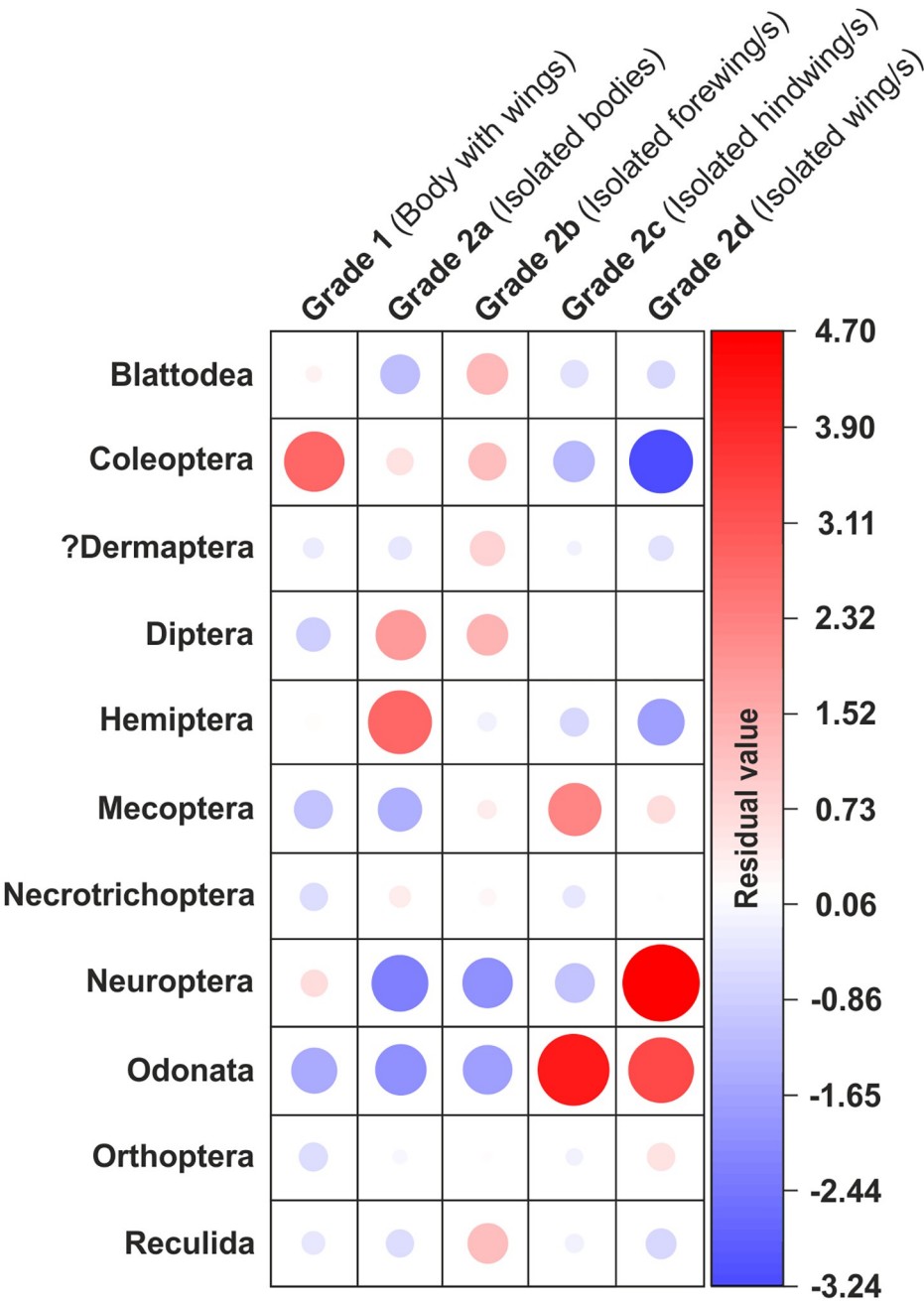

**Fig 9. Calculated Pearson residuals for each entry in the Chi-squared contingency table at order level, used to determine the degree to which each primary taphonomic grade contributed to the total Chi-squared score for the Alderton Hill palaeoentomofauna.** The colour of each circle represents its residual value: positive residuals are blue and negative residuals are red, indicating a positive and negative association respectively between taxonomy and taphonomy. The size of the circle is proportional to the degree of cell contribution to the relationship. Note: Ephemeroptera was not included in this analysis as the specimen attributed to this order (WARMS G 8084) was not available for re-examination and therefore could not be assigned a taphonomic grade.

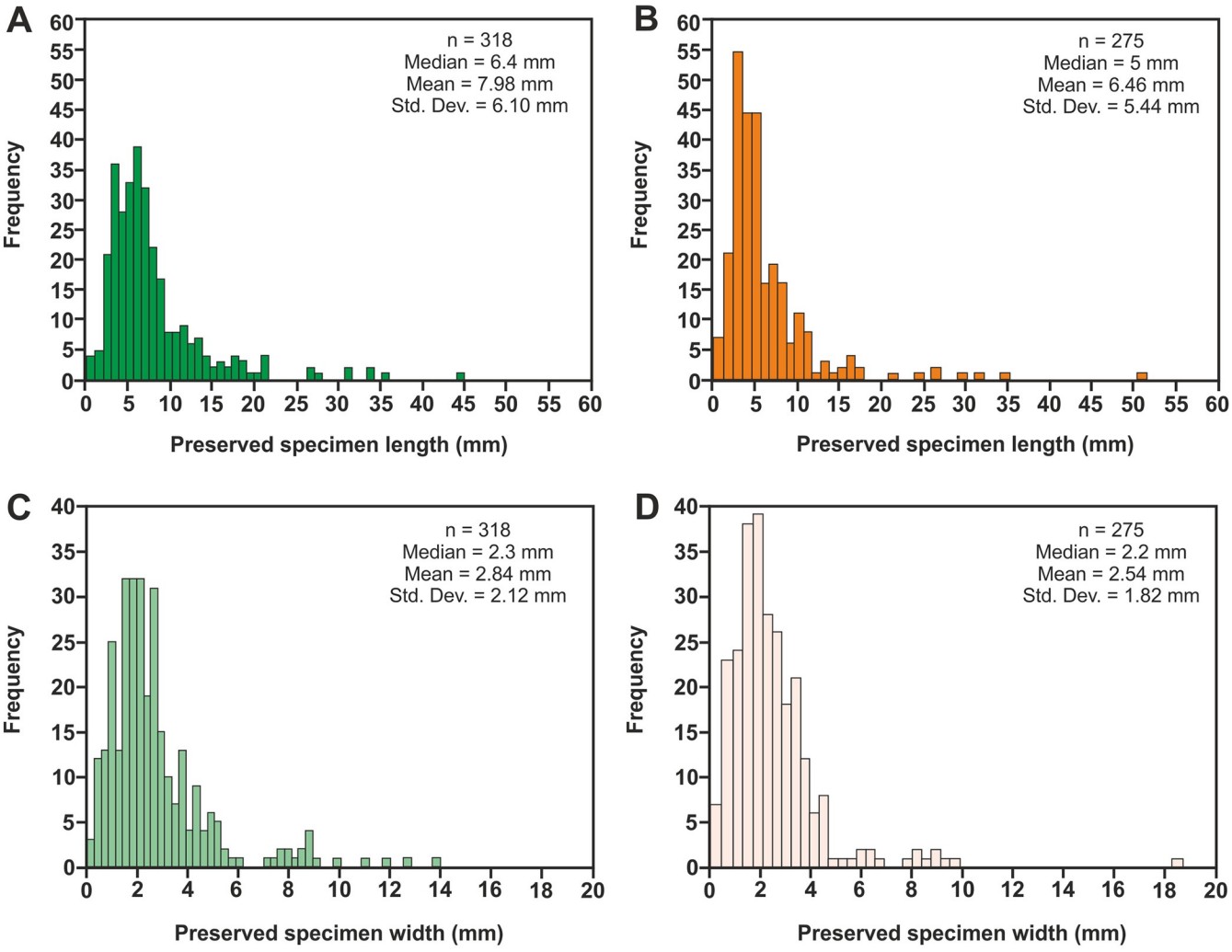

**Fig 10.** Preserved length and width of all insect elements recorded from: **(A, C)** Alderton Hill, Gloucestershire; **(B, D)** Strawberry Bank, Somerset.

The Strawberry Bank insects had a minimum and maximum preserved length of 0.8 mm and 51.2 mm respectively, with a median preserved length of 5 mm and a mean preserved length of 6.46 mm (SD = 5.44 mm; n = 275; Fig 10B). Preserved width ranged from 0.3 mm to 18.4 mm, with a median preserved width of 2.2 mm and a mean preserved width of 2.54 mm (SD = 1.82 mm; n = 275; Fig 10D). The distribution of the length and width shows a greater frequency of small sized elements at Strawberry Bank. Further statistical comparison of the size data from Strawberry Bank and Alderton Hill was not undertaken, as the taphonomic and taxonomic bias at Strawberry Bank render any results from this location meaningless.

## Discussion

The diversity of the Alderton Hill palaeoentomofauna, comprising 12 insect orders (Fig 7A), is comparable to the number of extant insect orders in present-day assemblages from analogous latitudes (30°– 40°N). For example, in the wetlands of the Zaranik Protectorate in North Sinai, Egypt (30.2° N), 11 insect orders occur [178]. Perhaps the most analogous present-day environment to Alderton Hill during the Toarcian is the Azores archipelago in the Atlantic. In

addition to the palaeolatitudinal similarity (35˚– 40˚N for Alderton Hill, and 37.7˚ N for the Azores), during the T-OAE, the Alderton Hill area had high rainfall and the closest land was a number of small islands similar to the Azores. Furthermore, these Toarcian islands were >1000 km from the nearest large continental landmasses of Laurentia and Eurasia (Fig 1B), this is comparable to the Azores that is *c.* 1400 km from mainland Europe. On the Azores, 19 insect orders occur (n = 5979) [179] of which 15 had evolved before the Toarcian and 10 of these are recorded within the Alderton Hill palaeoentomofauna. One order present at Alderton Hill, Mecoptera, is not documented from the Azores, and Reculida (also present at Alderton Hill) went extinct in the Jurassic. The similarity in order diversity at Alderton Hill, compared to present-day assemblages from analogous latitudes, suggests that the palaeoentomofauna is broadly representative of the number of orders you would expect to see in the Toarcian at this palaeolatitude.

Fossil insect collections can be biased towards better-preserved material with more diagnostic characters that allow the specimen to be readily identified e.g. isolated wings [180]. Based on our observations of the Alderton Hill assemblage, we consider that it is unlikely that there is a large collection bias for the following reasons: (1) the rarefaction curves show that the preserved insect assemblage is well-sampled at Alderton Hill at order level and reasonably well-sampled at family and genus levels (Fig 6); (2) the order diversity is comparable to insect assemblages at similar latitudes today; (3) the encompassing nodules are fine-grained and pale in colour, making it easy to visually distinguish the generally darker well-preserved fossils; (4) more delicate insect orders including Neuroptera and Diptera are well represented; (5) insect specimens in general are small and require further examination, meaning it is likely that all potential insect material was collected in the field; and (6) much of the material was not identified at the time of collection, again suggesting that all potential insect material that was found was collected.

High abundance and diversity of Odonata in present-day insect communities is regarded as an indication of good ecological health [181]. We therefore interpret the abundance (11.5%; n = 30) and diversity (represented by 3 families) of Odonata observed at Alderton Hill as indicative of a well-established and healthy insect ecosystem during the Toarcian. Hemiptera are not only the most diverse (4 families and 4 genera) order at Alderton Hill but are also the most abundant (22.1%; n = 58). Their abundance exceeds that in the present-day Azores, where they comprise *c.* 17% of the total insect assemblage [179]. We interpret the higher abundance and diversity of these herbivorous insects at Alderton Hill to be directly linked to the ready abundance of vegetation during the warm and mostly humid greenhouse climate of the T-OAE. In particular, the hemipteran family Progonocimicidae (Coleorrhyncha) identified within the Alderton Hill insect assemblage (NHMUK PI I.11465; Fig 3E) is characteristic of much warmer and humid conditions and would, most likely, have inhabited flora such as cheirolepid conifers [182], which were dominant during the T-OAE [38, 183]. Our data also supports the interpretation of Szwedo [51] who proposed that the Hemiptera, in particular the suborder Coleorrhyncha, diversified during the Jurassic in response to the changes in vegetation during the T-OAE. Similar to Hemiptera, the relatively high abundance of Orthoptera (13.4%; n = 35) within the Alderton Hill palaeoentomofauna, compared to 1% in the Azores today [179], is here attributed to their mostly herbivorous diet being well suited to the abundance of vegetation during the T-OAE. Coleoptera, whilst being the second most abundant order at Alderton Hill (20.6%; n = 54), are more common in the Azores today (32%) [179]. However, based on our current knowledge of this order, the high diversity within Coleoptera does not appear to have been achieved until the Late Jurassic *c.* 23 Ma later [111]; likely explaining the differences in dominance between the Toarcian and present-day.

The three most abundant orders at Alderton Hill (Hemiptera, Coleoptera and Orthoptera) are highly sclerotised taxa and together they comprise 56.1% (n = 147) of the total insect assemblage. This proportion is slightly elevated in comparison to the proportion of highly sclerotised taxa at present-day on the Azores archipelago, where Hemiptera, Coleoptera and Orthoptera comprise 50.4% (n = 3011) [179] of the total insect community across all of the islands. This could reflect one or more of the following for the Alderton Hill palaeoentomofauna:

(1) Higher abundance in the life assemblage because highly sclerotised taxa were better adapted to fluctuations in water, temperature and gases during the climatic extremes of the T-OAE. For instance, in Coleoptera, the elytra provide a mechanical barrier against direct water loss from the body surface, thereby increasing the organism's tolerance to desiccation [184] and enabling Coleoptera to withstand rapid temperature changes [185].

(2) Preferential preservation of highly sclerotised taxa and elements including elytra, hemelytra and tegmina. Decay studies of present-day insects [186, 187] have shown that highly sclerotised forewings are the strongest and most flexible elements, and therefore are more resistant to transportation and decay compared to those with less ossilszation. In contrast, less robust orders (e.g. soft-bodied Diptera) could have been preferentially removed during decay and decomposition.

(3) Alternatively, the concentration of highly sclerotised elements within the deposit could be due to predators, such as fish, marine reptiles or other insectivorous reptiles e.g. pterosaurs [74] and insectivorous mammals favouring consumption of the insect bodies over their wings; therefore leading to a concentration of highly sclerotised isolated wings [187, 188].

In contrast to Alderton Hill, the diversity of the Strawberry Bank palaeoentomofauna is lower as shown by comparison of the rarefaction curves for each taxonomic level, although at the family, genus and species levels, the rarefaction curves also show that this locality is incompletely sampled (Fig 6). Sclerotised taxa, specifically Coleoptera dominate the insect assemblage from Strawberry Bank (74.2%, n = 415). As this proportion is well in excess of present-day communities from similar palaeolatitudes, we interpret the Strawberry Bank insect assemblage to be a direct result of preferential preservation of sclerotised taxa. A preservational bias towards highly sclerotised orders at Strawberry Bank is further supported by: the greater frequency of small sized fragmentary elements (Fig 10); the poorer level of overall preservation, which for most of the specimens, prevents identification below order and family level; the greater proportions of isolated and poorly preserved wings (Fig 8B); and the fact that several less robust orders including Neuroptera and Diptera are well represented at Alderton Hill but are an order of magnitude lower in abundance at Strawberry Bank (Neuroptera: 1.1%, n = 6 at SB, compared to 12.2%, n = 32 at AH; Diptera: 0.2%, n = 1 at SB, compared to 9.5%, n = 25 at AH) (Fig 7). Williams *et al.* [41] suggested that the abundance of Coleoptera at Strawberry Bank reflected either genuinely higher abundance, collector bias or differential preservation. However, they went on to note that collector bias was unlikely as many of the insect specimens within the Moore collection were difficult to distinguish visually from the matrix, poorly preserved, and fragmentary, suggesting that Moore is more likely to have collected all of the insect material found in the field.

The poor preservation of the Strawberry Bank insects can be attributed to both the coarser-grained nature of the deposit [41] and their deposition nearer to shore where waves and current action was more prevalent than at Alderton Hill. We note however, that the fossil insect preservation is in contrast to other exceptionally well-preserved fauna collected from Strawberry Bank, including phosphatized soft tissue and three-dimensional preservation of marine nekton [89]. Although all of the Strawberry Bank fauna is reportedly from a single layer of nodules [41, 89], the difference in preservation may be because the insects are from a slightly

different stratigraphic level within the nodular horizon. In contrast to Strawberry Bank, the majority of insects found within the Alderton Hill insect assemblage are well-preserved, including wings showing intricate veining and evidence of colour pigmentation and patterns (Figs 3 – 5). These well-preserved wings allow for a high level of taxonomic identification, as demonstrated by the statistically significant relationship between taphonomic grade and taxonomic rank (Fig 9). Both lower energy conditions during deposition (as indicated by the sedimentology), and the clay-sized matrix (which allows for more delicate and detailed fossils to be preserved) are likely to be responsible for the better preservation of insects at Alderton Hill.

The level of disarticulation of the insect material from Alderton Hill (91.5%; Fig 8A) and Strawberry Bank (90.7%; Fig 8B) is commensurate with actualistic decay studies of present-day insects. For example, Duncan *et al.* [187], using cockroaches, determined that the average time required for complete mechanical disarticulation of an individual was 18 to 70 hours. The tegmina are the last elements of a cockroach to break down, and this occurs through the loss of the clavus in the anal region by separation along the wing venation [187]. Of the six cockroaches present within the assemblage, all but one of the tegmina from Alderton Hill are complete with the clavus (Fig 2G and Fig 2H), strongly suggesting that burial was likely to be rapid (< 70 hours). This provides further evidence that the fossil insects of Alderton Hill are well-preserved.

The sedimentology and the associated invertebrate fauna of the insect-bearing horizon of Alderton Hill and Strawberry Bank indicate that they were deposited within shallow-marine waters, with Strawberry Bank probably being closer to land. Whilst some insects may have been actively flying, or drifting on wood and plant matter over these shallow-marine areas, most of the insects from these localities must be allochthonous, having been transported from nearby land (Fig 1B). The allochthonous nature of these fossil assemblages may explain why no immature insect fossils have been found in the collections from Alderton Hill; however, Andrew J. Ross (pers. Comm.) did identify several possible insect larvae within the collections from Strawberry Bank. For many holometabolous insects (those undergoing complete metamorphosis), larval stages often possess few features that are robust enough to be preserved, although fossilised mouthparts have provided intriguing insights into the ecology of some species [189]. For some hemimetabolous insects (with incomplete metamorphosis) e.g. Hemiptera, juveniles can superficially look almost identical to adults, just smaller and without wings meaning that they might easily be misidentified as adults and therefore immature forms may be unrepresented within the current collections. Many hemimetabolous orders, however, do have very distinct larval stages, e.g. Odonata, and for this group the larval stages are not only robust enough to be preserved, but should also greatly outnumber adults for large parts of a given year (as they dominate the life cycle, e.g. some species have as many as 15 moults). Thus, immature stages may have been overlooked within the current historical collections from Alderton Hill and Strawberry Bank.

Whilst the exact reason(s) why only one stratigraphic level within the T-OAE contains an abundant and diverse assemblage awaits further study, we suggest a possible explanation for the high overall abundance of insects at Alderton Hill and Strawberry Bank is a specific event related to the highly unusual and extreme environmental conditions of the T-OAE. Periodic low oxygen conditions at Alderton Hill and most other epicontinental locations during the T-OAE are well established through a wide range of sedimentary, geochemical and biological proxies (see introduction). Noting the high abundance of the small early teleost fish *Leptolepis* and the larger pachycormids co-occurring with the insects at both localities [41, 64; new data set from this study, see Fig 2D], we postulate that this could be the direct result of shoals of fish being forced into shallower nearshore waters during seawater de-oxygenation, and dying in large numbers because of the lack of resources. The decaying floating fish would then have

become an attractive food source for predators, including Odonata and possibly other insects. Similar environmental events, often known as jubilees, are observed at present-day; in these cases, periods of intense de-oxygenation drive fish into surface-waters and shallow-water areas as they seek oxygenated water [190–193].

## Conclusions

Through the re-examination of all fossil insect collections from Alderton Hill, Gloucestershire, UK, we describe a rich and diverse palaeoentomofauna assigned to 12 insect orders, 21 families and 24 genera and 21 species. The quality of preservation allows for a high level of taxonomic identification, reflected by the locality having previously yielded 30 type specimens. The diversity of the Alderton Hill palaeoentomofauna at order level is comparable to present-day insect assemblages from equivalent latitudes (30˚– 40˚N), including the Azores archipelago, which is geographically and climatically interpreted to be most similar to the conditions at Alderton Hill during the Toarcian. This similarity in order diversity, in addition to the preservation of delicate taxa and the lack of a strong taxonomic bias lead us to interpret that the Alderton Hill palaeoentomofauna is a good overall reflection of the insect community during the T-OAE at this palaeolatitude.

The particularly high abundance of the Hemiptera and Orthoptera at Alderton Hill is likely to have been driven by dense vegetation on the neighbouring islands and landmasses, which would have thrived in the warm humid conditions of the T-OAE. The high diversity of Hemiptera present at Alderton Hill supports the interpretation that changes in the vegetation in response to climate fluctuations may have driven the evolution of this order. A slightly higher proportion of sclerotised taxa in the Alderton Hill assemblage compared to present-day could reflect that their less permeable exoskeletons provided greater tolerance to fluctuations in fluids and temperature at this time. Alternatively, there could be a minor taphonomic bias towards more robust taxa and elements, or preferential predation of insect bodies.

In comparison to Alderton Hill, the coeval UK insect assemblage of Strawberry Bank, Somerset is less diverse with 2 superorders, 9 insect orders, 12 families, 6 genera and 3 species, and is heavily dominated by Coleoptera elytra. This taxonomic and taphonomic bias is despite the fact that more insect specimens have been collected from Strawberry Bank (n = 799) than at Alderton Hill (n = 370).

The dominance of isolated wings and wing fragments, in addition to the high levels of disarticulation amongst the insect material from both Alderton Hill and Strawberry Bank is consistent with actualistic decay studies of present-day insects. The greater proportion of smaller insect elements at Strawberry Bank compared to Alderton Hill reflects greater fragmentation and supports the interpretation of taphonomic bias within the Strawberry Bank material. In contrast to the poorly preserved insects of Strawberry Bank, the majority of the wings within the Alderton Hill insect assemblage are well-preserved, with intricate veining and colour pigmentation patterns.

We propose that the co-occurrence of fish and insects within this interval of the T-OAE is related to environmental conditions during the T-OAE. A possible scenario is that shoals of fish were forced into shallower, near shore waters during intense de-oxygenation events, where they died and became a food source for higher trophic predators, including insects.

Our results indicate that the fossil insects from Alderton Hill represent a well-preserved and diverse assemblage that provides a useful insight into the palaeoentomofauna at temperate latitudes during the T-OAE. Comparison with the coeval assemblage of Strawberry Bank shows that the Alderton Hill insect assemblage can currently be regarded as the best-preserved and most representative record of Toarcian insects in the UK.

## Acknowledgments

This research was supported by NERC CENTA (Central England NERC Training Alliance) PhD studentship to ES [grant number NE/S007350/1]. The authors acknowledge the following museums and institutions for allowing access and facilitating the examination of the Alderton Hill insect material and/or for providing relevant specimen information and photographs included within this publication: BGS (Simon Harris and Paul Shepherd); BRSMG (Deborah Hutchinson); CHAGM (Benedict Sayers and Eleanor Edwardes); CAMSM (Elizabeth Harper and Matt Riley); NHMUK (Rich Howard); TTNCM (Amal Khreisheh); WARMS (Jon Radley). Thanks to Mike Simms for early discussion and Ulysses Thomson (né Richard Kelly) for his assistance with locating Lower Jurassic insect collections at the beginning of this project. Matt Williams (Bath Royal Literary and Scientific Institution) and Andrew J. Ross (National Museum of Scotland) very kindly provided their database of the Strawberry Bank material at TTNCM, which was of assistance in the compilation of our own extended database for Strawberry Bank. Matthew Butler is thanked for assisting in locating, cataloguing and photographing the associated fauna of Alderton Hill within the NHMUK collections. Many thanks to André Nel, Andrew J. Ross and an anonymous reviewer for their suggestions that helped to improve this manuscript.

JA contributed only to the catalogue of identified taxa in this study and does not wish to endorse the interpretation of the data in the Discussion section.

No permits were required for the described study, which complied with all relevant regulations.

## Author Contributions

**Conceptualization:** Emily J. Swaby, Angela L. Coe, Bryony A. Caswell.

**Data curation:** Emily J. Swaby, Liadan G. Stevens.

**Formal analysis:** Emily J. Swaby, Angela L. Coe, Luke Mander.

**Funding acquisition:** Angela L. Coe, Bryony A. Caswell, Scott A. L. Hayward, Luke Mander.

**Investigation:** Emily J. Swaby, Angela L. Coe, Jörg Ansorge.

**Supervision:** Angela L. Coe, Bryony A. Caswell, Scott A. L. Hayward, Luke Mander.

**Visualization:** Emily J. Swaby, Aimee McArdle.

**Writing – original draft:** Emily J. Swaby, Angela L. Coe.

**Writing – review & editing:** Emily J. Swaby, Angela L. Coe, Jörg Ansorge, Bryony A. Caswell, Scott A. L. Hayward, Luke Mander, Liadan G. Stevens.

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
