## [Decision Letter · Decision Letter 0]

13 Nov 2023

PONE-D-23-33017The fossil insect assemblage associated with the Toarcian (Lower Jurassic) Oceanic Anoxic Event from Alderton Hill, Gloucestershire, UKPLOS ONE

Dear Dr. Swaby,

Thank you for submitting your manuscript to PLOS ONE. After careful consideration, we feel that it has merit but does not fully meet PLOS ONE’s publication criteria as it currently stands. Therefore, we invite you to submit a revised version of the manuscript that addresses the points raised during the review process.

We look forward to receiving your revised manuscript.

Kind regards,

Miquel Vall-llosera Camps

Senior Staff Editor

PLOS ONE

Journal Requirements:

2. In your manuscript, please provide additional information regarding the specimens used in your study. Ensure that you have reported human remain specimen numbers and complete repository information, including museum name and geographic location. 

For more information on PLOS ONE's requirements for paleontology and archeology research, see https://journals.plos.org/plosone/s/submission-guidelines#loc-paleontology-and-archaeology-research.

4. We note that Figure 1 in your submission contain map/satellite images which may be copyrighted. All PLOS content is published under the Creative Commons Attribution License (CC BY 4.0), which means that the manuscript, images, and Supporting Information files will be freely available online, and any third party is permitted to access, download, copy, distribute, and use these materials in any way, even commercially, with proper attribution. For these reasons, we cannot publish previously copyrighted maps or satellite images created using proprietary data, such as Google software (Google Maps, Street View, and Earth). For more information, see our copyright guidelines: http://journals.plos.org/plosone/s/licenses-and-copyright.

Reviewers' comments:

Reviewer's Responses to Questions

**Comments to the Author**

1. Is the manuscript technically sound, and do the data support the conclusions?

Reviewer #1: Yes

Reviewer #2: Yes

2. Has the statistical analysis been performed appropriately and rigorously? 

Reviewer #1: No

Reviewer #2: Yes

3. Have the authors made all data underlying the findings in their manuscript fully available?

Reviewer #1: Yes

Reviewer #2: Yes

4. Is the manuscript presented in an intelligible fashion and written in standard English?

Reviewer #1: Yes

Reviewer #2: Yes

5. Review Comments to the Author

Reviewer #1: only assemblages from Germany have been studied in detail

in Luxembourg too

Hemiptera, Coleoptera and Orthoptera are the commonest

indeed, in extant assemblages, orthoptera are not so commmon, less than diptera and hymenoptera, thus the sentence juts before is problematic, maybe ?

the three main collections, although rather large, are not so important compared to the potential diversity of insects, that the percentages as given, can be taken without caution

that is, if, it was possible, to make random sub-collections within these collections, to establish at least mean values and rudimentary statistical information for each order. This would increase the significance of the percentages that you use

as I have seen in many other outcrops, collections of thousands of insect fossils need tobe made in order to have rather stable percentages of the orders

as you base all your study on these percentages, it would better if you could make such statistical analyses

otherwise the taxonomy is ok

the figures are ok

references ok

english quite ok

clarity ok

Reviewer #2: It is a very interesting and important contribution to the knowledge of the Toarcian strata of UK and Europe. It is based on rich material and presents very good interpretation of data acquired during the investigations. Very interesting and important part of the work is taphonomic analysis, it is very well presented. A few minor changes are necessary, improvement, e.g. use of German word Lagerstätte, should be in majuscule first letter as German noun, in the Jurassic most of the lineages mentioned were specialized forms so use of 'primitive' in context of old insect lineages seems to be awkward; there are some minor typographic errors, etc. The main concern of the manuscript is usage of scientific names. Names in full with author(s) and year of description should be given with comma separating Author(s) name and year - it is written in Article 22 of the ICZN Code and also pointed in Recommendations of ICZN. This should be checked and improved throughout the text. Auhots used expression "previous holotype", which is very odd - a holotype of species, even if the name is synonymized remains a holotype - this expression should be changed throughout the text.

If 'Elcana' is no longer available as generic name (put in synonymy), the family name 'Elcanidae' probably should be changed or conserved - is any decision of ICZN commission made? I cannot find any note on it.

If Elcana is synonymized under Panorpidium, these nomenclatorial combinations should be used. Please make these decisions clear, if you resurrecting Elcana from synonymy for a few species? The you have to give the concept and content of the taxon under this name.

If you decided to use the tradional name Auchenorrhyncha (which is a paraphyletic assemblage), please give the reasons for this treatment.

Fulgoridiidae (another paraphyletic assemblage) were recently treated by Poinar et al. (2022), Bourgoin & Szwedo (2022, 2023), Zhao et al. (2023), so please refer to these works for discussion.

Please note that Handlirsch monograph appeared in sections 1906 to 1908, and the section dealing with Jurassic Fulgoridiidae appeared in 1906, not in 1908 as suggested by dates associated with taxonomic names. The detailed dates of appearnce of particular Lieferungen were presented e.g. in Szwedo et al. (2004) Catalogue

Taxonomic actions presented in theses are not valid according to rules of ICZN - if you agree with these decisions, it must be clearly stated in your paper and you are reviewers of these taxa.

6. PLOS authors have the option to publish the peer review history of their article (what does this mean?). If published, this will include your full peer review and any attached files.

Reviewer #1: **Yes: **andre nel

Reviewer #2: No

---

## [Author Response · Author response to Decision Letter 0]

19 Dec 2023

This is all outlined in the submitted document 'Swaby et al. Response to Reviewers'.

---

## [Decision Letter · Decision Letter 1]

23 Jan 2024

PONE-D-23-33017R1The fossil insect assemblage associated with the Toarcian (Lower Jurassic) Oceanic Anoxic Event from Alderton Hill, Gloucestershire, UKPLOS ONE

Dear Dr. Swaby,

Thank you for submitting your manuscript to PLOS ONE. After careful consideration, we feel that it has merit but does not fully meet PLOS ONE’s publication criteria as it currently stands. Therefore, we invite you to submit a revised version of the manuscript that addresses the points raised during the review process.

We look forward to receiving your revised manuscript.

Kind regards,

Ben Thuy

Academic Editor

PLOS ONE

Journal Requirements:

**Additional Editor Comments:**

Your revised manuscript has now been reviewed by another referee (reviewer 3). It is my pleasure to let you know that reviewer 3 has recommended publication of your manuscript in PLOS ONE, provided that you address some minor issues. I invite you to revise your manuscript taking into account the comments by reviewer 3. Should you have any questions, do not hesitate to get in touch. I look forward to receiving your revised manuscript.

Reviewers' comments:

Reviewer's Responses to Questions

**Comments to the Author**

1. If the authors have adequately addressed your comments raised in a previous round of review and you feel that this manuscript is now acceptable for publication, you may indicate that here to bypass the “Comments to the Author” section, enter your conflict of interest statement in the “Confidential to Editor” section, and submit your "Accept" recommendation.

Reviewer #1: All comments have been addressed

Reviewer #3: (No Response)

2. Is the manuscript technically sound, and do the data support the conclusions?

Reviewer #1: Yes

Reviewer #3: Yes

3. Has the statistical analysis been performed appropriately and rigorously? 

Reviewer #1: Yes

Reviewer #3: Yes

4. Have the authors made all data underlying the findings in their manuscript fully available?

Reviewer #1: Yes

Reviewer #3: Yes

5. Is the manuscript presented in an intelligible fashion and written in standard English?

Reviewer #1: Yes

Reviewer #3: Yes

6. Review Comments to the Author

Reviewer #1: all comments have been adressed

just a remark for future researches on these insects

photograph under uv light works quite well for such insects, as we experiment here

nice paper

congratulation

andré

Reviewer #3: Review of Swaby et al. ‘Fossil insects associated with the Toarcian Oceanic Anoxic Event from Alderton Hill, UK’

This review is of a revised manuscript that was reviewed by others previously.

The paper is a thorough assessment of the Toarcian insect fauna of the UK and deserves to be published. However, I have spotted some things that could be improved, additional references that would be worth citing and some corrections, thus I am recommending minor revision.

Line 97, ref. 68 is 20th Century.

Line 107, the thylacocephalans from Strawberry Bank were mentioned by-

Laville, T., Forel, M.-B. & Charbonnier, S. 2023. Re-appraisal of thylacocephalans (Euarthropoda, Thylacocephala) from the Jurassic La Voulte-sur-Rhône Lagerstätte. European Journal of Taxonomy, 898, 1-61.

Lines 161, 176, ‘Eryon’ richardsoni is now in the genus Proeryon, see-

Audo, D., Schweigert, G. & Charbonnier, S. 2020. Proeryon, a geographically and stratigraphically widespread genus of polychelidan lobsters. Annales de Paléontologie, 106, 102376, 1-21.

Line 219, in between ‘collated’ and ‘by’ please insert ‘studied by Andrew J. Ross (National Museums Scotland), a list of preliminary identifications was supplied to the senior author and the fauna was summarised‘ [I spent a significant amount of time identifying these, so deserve more credit]

Line 246, in between ‘by’ and ‘Williams’ insert ‘Ross in’

Lines 293-294, replace ‘incertae sedis’ with ‘undescribed’

Line 298, it would be worth mentioning that M is a leg.

Line 308, H looks like it is good enough to be identified or described so I suggest you replace ‘incertae sedis’ with ‘undetermined’

Line 370 onwards. This large section entitled ‘Systematic taxonomy’ is not satisfactory because it does not include any detailed descriptions, nor does it mention the diagnostic characters that justify the identification of the specimens. This is little more than a catalogue of identified taxa with comments, so should be re-titled accordingly.

Lines 502-510, this is being described elsewhere (paper in prep.) so replace ‘incertae sedis’ with ‘uncertain’ and ‘gen. et sp. nov.’ with ‘undescribed’ as it is not being named here. Replace the last sentence with ‘The family placement of this specimen is uncertain at present.’

Lines 529, 972, additional references are-

Zessin W. 1983. Revision der mesozoischen Familie Locustopsidae unter Berücksichtigung neuer Funde (Orthopteroida, Caelifera). Deutsche Entomologische Zeitschrift, 30, 173–237.

Gorochov, A.V. & Coram, R.A. 2023. New and little known taxa of the order Orthoptera (Insecta) from the Upper Triassic and Lower Jurassic of England. 2023. Palaeoentomology, 6(2), 198-204.

Lines 550-553, this is not sufficient for formally synonymising these species. You need to explain in detail the characters that justify the synonymy and provide a synonymy list with the term ‘syn. nov.’ Given that this is not a detailed systematic review of ‘Elcana’ I suggest you replace the last sentence with ‘However, these species may prove to be junior synonyms.’

Line 592, I.11405 (Figure 3M) is a leg.

Lines 597-599, only wings are mentioned, what about isolated legs?

Line 732-3, what character and what about it justifies rejecting Austalius from Necrotauliidae?

Line 735, A recent paper on necrotauliids that should be added is-

Sukatsheva, I.D. & Sinitshenkova, N.D. 2023. New caddisflies (Insecta: Trichoptera, Necrotauliidae, Philopotamidae) from the Jurassic of Asia and their Triassic ancestors. Paleontological Journal, 57(5), 529-540.

Line 787, replace ‘as’ with ‘a’

Line 881, move ‘Assigned…’ to the next line.

Line 919, ‘Strawberry Bank entomofauna’ should not be in italics.

There is a lot of repetition of the taxonomic names and references as used for Alderton Hill above. I consider it would be better to combine both faunas and then under ‘Assigned material’ make it clear which specimens are from Alderton Hill and which are from Strawberry Bank.

Line 990, add ref. 141.

Line 996, ref. 41 does not mention Protogryllidae.

Line 1025, replace ‘these specimens’ with ‘this specimen’

Line 1135, add ref. 41.

Line 1342, this specimen was examined relatively recently by Thomson (Kelly) for his PhD, so I really hope it is not lost. It is much more likely that it has been misplaced so I suggest you delete ‘and is presumed lost.’

Line 1501, add insectivorous mammals

Line 1529, it seems more likely that the insects came from a tabular limestone; as far I recall none of the specimens have curved surfaces that would indicate they came from nodules.

Lines 1556, 1566, I identified several possible insect larvae from Strawberry Bank (see my preliminary list).

Andrew Ross

7. PLOS authors have the option to publish the peer review history of their article (what does this mean?). If published, this will include your full peer review and any attached files.

Reviewer #1: **Yes: **andre nel

Reviewer #3: No

---

## [Author Response · Author response to Decision Letter 1]

5 Feb 2024

Please see the uploaded 'Swaby et al. Response to Reviewers' for our response to specific reviewer and editor comments.

---

## [Editor Report · Decision Letter 2]

13 Feb 2024

The fossil insect assemblage associated with the Toarcian (Lower Jurassic) Oceanic Anoxic Event from Alderton Hill, Gloucestershire, UK

PONE-D-23-33017R2

Dear Ms. Swaby,

We’re pleased to inform you that your manuscript has been judged scientifically suitable for publication and will be formally accepted for publication once it meets all outstanding technical requirements.

Kind regards,

Ben Thuy

Academic Editor

PLOS ONE

---

## [Editor Report · Acceptance letter]

26 Feb 2024

PONE-D-23-33017R2 

PLOS ONE

Dear Dr. Swaby, 

I'm pleased to inform you that your manuscript has been deemed suitable for publication in PLOS ONE. Congratulations! Your manuscript is now being handed over to our production team.

Kind regards, 

on behalf of

Dr. Ben Thuy 

Academic Editor

PLOS ONE